# An all-in-one pipeline for the in vitro discovery and in vivo testing of *Plasmodium falciparum* malaria transmission blocking drugs

Nicolas M. B. Brancucci [1,2] ✉, Christin Gumpp[1,2], Geert-Jan van Gemert[3], Xiao Yu[1,2], Armin Passecker[1,2], Flore Nardella[4,8], Basil T. Thommen [1,2,10], Marc Chambon[5], Gerardo Turcatti [5], Ludovic Halby[6], Benjamin Blasco[7,9], Maëlle Duffey [7,9], Paola B. Arimondo[6], Teun Bousema [3], Artur Scherf [4], Didier Leroy [7], Taco W. A. Kooij [3], Matthias Rottmann [1,2] ✉ & Till S. Voss [1,2] ✉

Elimination of malaria will require new drugs with potent activity against *Plasmodium falciparum* mature stage V gametocytes, the only stages infective to the mosquito vector. The identification and comprehensive validation of molecules active against these quiescent stages is difficult due to the specific biology of gametocytes, challenges linked to their cultivation in vitro and the lack of animal models suitable for evaluating the transmission-blocking potential of drug candidates in vivo. Here, we present a transmission-blocking drug discovery and development platform that builds on transgenic NF54/iGP1_RE9H^ulg8 parasites engineered to conditionally produce large numbers of stage V gametocytes expressing a red-shifted firefly luciferase viability reporter. Besides developing a robust in vitro screening assay for the reliable identification of stage V gametocytocidal compounds, we also establish a preclinical in vivo malaria transmission model based on infecting female humanized NODscidIL2Rγ^null mice with pure NF54/iGP1_RE9H^ulg8 stage V gametocytes. Using whole animal bioluminescence imaging, we assess the in vivo gametocyte killing and clearance kinetics of antimalarial reference drugs and clinical drug candidates and identify markedly different pharmacodynamic response profiles. Finally, we combine this mouse model with mosquito feeding assays and thus firmly establish a valuable tool for the systematic in vivo evaluation of transmission-blocking drug efficacy.

Half of the world's population lives at risk of contracting malaria, a devastating infectious disease caused by protozoan parasites of the genus *Plasmodium* and transmitted by female *Anopheles* spp. mosquitoes. Over 95% of the global 249 million malaria cases and 608,000 deaths reported for 2022 were caused by *P. falciparum*[1]. Upon injection of sporozoites into the skin via an infectious mosquito bite and a first round of parasite multiplication inside hepatocytes, *P. falciparum* colonizes the human blood. Here, merozoites invade red blood cells

---

(RBCs) and develop intracellularly through the ring and trophozoite stage before mitotic replication via schizogony produces up to 32 daughter merozoites that egress from the infected RBC (iRBC) to invade new erythrocytes. Continued repetition of these 48-hour replication cycles is responsible for all malaria symptoms and chronic infection. At the same time, a small proportion of parasites produced during each round of replication commit to gametocytogenesis via an epigenetic switch that activates expression of AP2-G, the master transcriptional regulator of sexual conversion[2]. These parasites produce sexual ring stage progeny that differentiate within ten to twelve days and across five morphologically distinct stages into either female or male stage V gametocytes, the only forms able to infect mosquitoes. While stage I-IV gametocytes sequester away from circulation, primarily in the bone marrow and spleen parenchyma, stage V gametocytes are released back into the bloodstream where they circulate as quiescent cells for up to several weeks[3]. Once ingested by a mosquito, female and male gametocytes are rapidly activated to release one macrogamete and eight flagellated microgametes, respectively. After fertilization, ookinete transformation and oocyst development, thousands of sporozoites eventually colonize the salivary glands for onward transmission to another human host.

The emergence and spread of insecticide-resistant mosquitoes and parasite strains partially resistant to first-line artemisinin-based combination therapies (ACTs) threaten the malaria elimination and eradication agenda[4,5]. An additional major shortcoming of current antimalarials is their poor activity against gametocytes. Some drugs are inactive against all gametocyte stages (e.g. pyrimethamine) and while others are active against immature stage I-III (e.g. choloroquine) or even stage I-IV gametocytes (e.g. artemisinin and derivatives), almost all fail to kill quiescent stage V gametocytes[6-10]. As a consequence, infected individuals can remain infectious to mosquitoes for weeks even after asexual parasites and immature gametocytes have been cleared by drug treatment[11-13]. Primaquine (PQ), the only licensed drug with potent in vivo gametocytocidal and transmission-blocking activity, can cause severe hemolysis in people with glucose-6-phosphate dehydrogenase (G6PD) deficiency, a human genetic disorder frequently observed in malaria-endemic regions[14,15]. ACTs combined with a single low-dose of PQ of 0.25 mg/kg given on the first day of ACT treatment, as recommended by the WHO as a strategy to reduce malaria transmission[16], have been shown to be safe and effective in preventing mosquito infection in the field[17-20] but are not yet widely implemented. Furthermore, while artemether-lumefantrine, but not other ACTs, exerts potent transmission-reducing activity even in the absence of PQ[21,22], there is growing concern about the spread of partial artemisinin resistance on the African continent[23,24]. This concern is fueled by indications that parasites with partial resistance to artemisinin are more likely to present gametocytes[25] and may have a transmission advantage under artemisinin drug pressure[26].

To guide the discovery and development of urgently needed new antimalarial drugs, the Medicines for Malaria Venture (MMV) developed target candidate profiles (TCPs) describing the desired minimal and ideal properties of molecules to be combined in next generation medicines (target product profiles, TPPs)[8]. In addition to the strict requirement for drugs with potent activity against the disease-causing asexual blood stage parasites and novel mode-of-action to overcome drug resistance (TCP-1), transmission-blocking activity (TCP-5) is considered an essential property of new products for malaria elimination. Effective transmission-blocking drug combinations must have potent activity against asexual parasites and all gametocyte stages and can contain dual-active compounds or gametocyte-clearing compounds combined with molecules active against asexual blood stages. Although dual-active molecules are preferred from a drug development perspective, they are likely to lose their transmission-blocking effectiveness once drug resistance emerges, a risk much less likely to occur for compounds that specifically target the non-proliferative gametocyte stages[7].

Due to the highly specialized biology of gametocytes, the development of suitable gametocytocidal drug assays is not straightforward. Gametocytes typically emerge at low frequency (<10%) from the pool of asexual parasites and require twelve days for full maturation[2]. During this period, gametocytes undergo dramatic cellular and metabolic transformations, culminating in the temporary quiescence of mature stage V gametocytes, while asexual parasites continue to replicate and produce new cohorts of gametocytes every second day[27]. One major challenge in assay development, therefore, lies in the need to produce pure and synchronous gametocyte populations for an accurate assessment of sexual stage-specific compound activity, and in large enough numbers to facilitate the screening of chemical libraries. Another challenge is that assay readouts capable of measuring cell viability, rather than proliferation, are required. Several laboratories addressed these challenges, and a multitude of different assay formats has been developed. Current protocols for gametocyte production apply conditions of high asexual parasitemia as a stress factor to enhance sexual commitment rates (SCRs). One approach employs daily medium changes without addition of fresh RBCs, which leads to overgrowth and death of asexual parasites while emerging gametocytes continue to mature with a rather poor level of synchronicity[28-30]. Alternatively, parasites are exposed to conditioned medium (i.e. the supernatant of high parasitemia cultures), followed by treating the progeny with N-acetyl-glucosamine (GlcNAc) to eliminate asexual parasites, which delivers synchronous gametocytes but relies on complicated culture treatment protocols[31-34]. Both approaches achieve SCRs rarely exceeding 30%, require large volumes of asexual feeder cultures, and many protocols rely on gametocyte enrichment by gradient centrifugation[35-38] or magnetic capture[6,9,39,40]. Methods applied to assess gametocyte viability are based for instance on quantifying parasite metabolic[10,35,37,41-47], enzymatic[48,49] or mitochondrial activity[6,50,51], luciferase reporter enzyme activity[9,10,40,46,52-57] or gamete formation[36,38,39,58,59]. Regardless of the type of assay used, hit compounds must be further tested in the Standard Membrane Feeding Assay (SMFA), where mosquitoes feed on compound-exposed gametocytes[60,61], to confirm effective transmission-blocking activity[7,28].

The MMV Malaria Box, a selection of 400 molecules identified as active against asexual blood stage parasites in high-throughput screening (HTS) campaigns[62], has been tested against gametocytes on different assay platforms. These studies highlighted that inter-assay differences in gametocyte culture and purification protocols, level of gametocyte synchronicity, stage composition at the time of compound exposure, compound exposure times and/or the methods used to quantify gametocyte viability strongly impact assay outcome. Consequently, only about half of the gametocytocidal molecules reported showed consistent activity across multiple assays[6,38-40,50,53,54,59,63,64]. When considering hits identified at least twice independently, about 60 MMV Malaria Box compounds show low- to sub-micromolar activity against immature gametocytes (stage I-IV). Similar to antimalarial drugs, most of these compounds are more potent against early-stage gametocytes and lose their efficacy once gametocytes transitioned into their final stage of maturation[6,39,59], underscoring the importance of performing primary screens on quiescent stage V gametocytes. A small number of compounds with potent activity against mature stage V gametocytes were still identified (e.g. MMV019918, MMV665941), and these hits are also active against immature stages and block parasite transmission to mosquitoes in SMFAs[6,39,59,65]. Several hundred additional dual-active molecules have been identified through the screening of over 17,000 experimental antimalarial compounds represented in the TCAMS and GNF Malaria Box libraries[6,42,66]. The screening of compound collections consisting of approved drugs, clinical drug candidates and pharmacologically active molecules[37,63], inhibitors of human kinases[63] or epigenetic

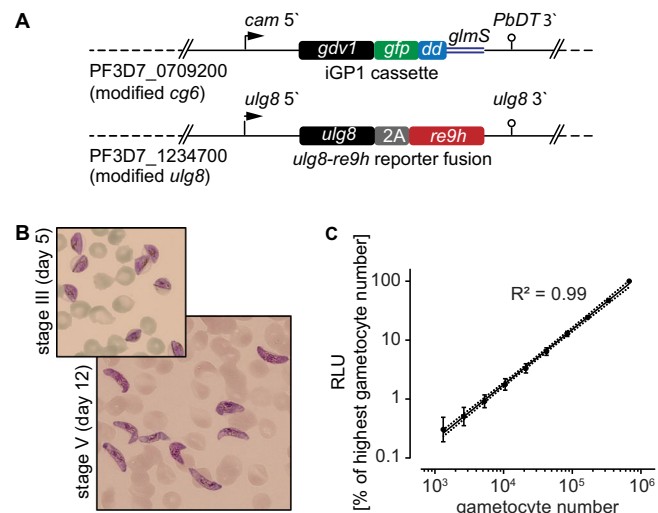

**Fig. 1 | NF54/iGP1_RE9H^ulg8 gametocyte reporter line.** Engineering and validation of the NF54/iGP1_RE9H^ulg8 reporter line for gametocytocidal drug research. **A** Schematic maps of the genetically modified loci in NF54/iGP1_RE9H^ulg8 parasites. The *cg6* locus carries an integrated conditional GDV1-GFP-DD over-expression cassette (iGP1) controlled by the *P. falciparum* calmodulin promoter (cam 5′) and the glmS ribozyme element followed by the *P. berghei* dihydrofolate-thymidylate synthase terminator (PbDT 3′)[71]. The *ulg8* gene is tagged at the 3′ end with a sequence encoding the 2 A split peptide and the RE9H luciferase. **B** Hemacolor-stained thin blood smears showing stage III (day 5) and mature stage V (day 12) NF54/iGP1_RE9H^ulg8 gametocytes. Representative images of one of n = 4 biological replicates are shown. **C** Correlation between NF54/iGP1_RE9H^ulg8 stage V gametocyte numbers (day 12) and RE9H-catalysed bioluminescence. Values on the y-axis represent RLUs normalized to the mean signal emitted from the wells containing the highest gametocyte number, obtained from n = 4 biological replicates (mean ± s.d.). Linear regression (black line), coefficient of determination (R²) and confidence bands (dashed lines) are indicated.

regulators[67,68], as well as HTS campaigns testing chemical diversity libraries totaling over 400,000 molecules[6,47,52,53,69,70] identified no more than a few hundred compounds with promising activity against gametocytes, highlighting an extreme sparsity of gametocyte-targeting hits among molecules that have not been prescreened against asexual blood stage parasites. Interestingly, some of these hits show higher potency against gametocytes compared to asexual parasites, suggesting these molecules may target biological pathways/processes upregulated in or even specific to gametocytes[6,46,52,69]. Together, these studies identified numerous promising chemical starting points for the development of transmission-blocking drug candidates. However, since many of the above screening efforts used semi-synchronous late-stage gametocytes at the time of compound exposure, it remains unclear how many of the reported hits retain potent activity against quiescent stage V gametocytes.

Future efforts in transmission-blocking drug discovery and development will profit from improved and new methodologies overcoming the limitations of currently applied assays. Robust protocols for the cost-effective and efficient production of synchronous gametocytes will be important to allow better standardization of assays for stage-specific compound activity profiling and the screening of chemical libraries against mature stage V gametocytes. Furthermore, a preclinical model for gametocytocidal and transmission-blocking drug efficacy testing in vivo is urgently required. To tackle these challenges, we advanced our recently published NF54/iGP1 parasite line that allows for the mass production of synchronous gametocytes via controlled overexpression of gametocyte development 1 (GDV1)[71], an upstream activator of AP2-G expression[72]. We engineered NF54/iGP1-RE9H^ulg8 parasites expressing a red-shifted firefly luciferase specifically in gametocytes and established a simple

and reliable in vitro luminescence-based stage-specific gametocyte viability assay. The screening of four chemical libraries validated this assay as a suitable tool for stage V gametocytocidal drug discovery. Most importantly, we also utilized NF54/iGP1-RE9H^ulg8 stage V gametocytes to develop a NODscidIL2Rγ^null (NSG) mouse model for *P. falciparum* transmission and applied this model to evaluate the in vivo gametocytocidal and transmission-blocking activities of antimalarial drugs and clinical drugs candidates.

## Results

### Engineering of NF54/iGP1_RE9H^ulg8 parasites

NF54/iGP1 parasites carry a double conditional GDV1-GFP-DD-glmS expression cassette integrated into the non-essential *cg6* locus (PF3D7_0709200)[71,73]. When cultured in the presence of 2.5 mM D-(+)-glucosamine hydrochloride (GlcN) and the absence of Shield-1 (+ GlcN/−Shield-1), the glmS riboswitch element in the 3′ untranslated region mediates *gdv1-gfp-dd* mRNA degradation and the C-terminal FKBP destabilisation domain (DD) mediates GDV1-GFP-DD protein degradation[74,75]. Under these conditions, NF54/iGP1 parasites display baseline sexual conversion rates (SCRs) of approximately 8%[71]. When GlcN is removed and 1.35 μM Shield-1 added for 48 h to a synchronous population of ring stage parasites (−GlcN/ + Shield-1), GDV1-GFP-DD is stably expressed and up to 75% of schizonts commit to sexual development, producing sexual ring stage progeny that differentiate in a synchronous manner into mature stage V gametocytes[71].

Here, we engineered NF54/iGP1 parasites expressing the ATP-dependent red-shifted *Photinus pyralis* firefly luciferase variant PpyRE9H (RE9H)[76] under control of the endogenous gametocyte-specific *ulg8* (upregulated in late gametocytes 8) promoter (PF3D7_1234700)[77] (Fig. 1A and Fig. S1). RE9H-mediated oxidation of the D-luciferin substrate produces light at a peak emission wavelength of 617 nm (compared to 562 nm for wild type firefly luciferase)[76]. Because light above 600 nm has improved tissue penetration properties, red-shifted luciferases are preferred reporters for in vivo bioluminescence imaging[78,79], and RE9H has successfully been used for the sensitive detection of *Trypanosoma cruzi* and *T. brucei* in experimental mouse infections models[80,81]. The *ulg8* upstream and downstream regions have previously been shown to regulate the specific and robust expression of GFP and luciferase reporters in both female and male gametocytes, with increased expression observed in late-stage gametocytes[77]. Hence, we used the CRISPR/Cas9 system to tag the *ulg8* gene in NF54/iGP1 parasites in frame with a sequence encoding the 2 A split peptide[82,83] fused to RE9H (Fig. 1A and Fig. S1). After drug selection of transgenic parasites, two clonal NF54/iGP1_RE9H^ulg8 populations were obtained by limiting dilution cloning. PCRs on genomic DNA (gDNA) confirmed correct editing of the *ulg8* locus and NF54/iGP1_RE9H^ulg8 clone B2 (hereafter termed NF54/iGP1_RE9H^ulg8) was selected for further experiments (Fig. S1).

NF54/iGP1_RE9H^ulg8 asexual parasites replicated with the same efficiency (multiplication rate of 7.0 ± 0.3 s.d.) compared to NF54 wild type parasites (6.9 ± 0.8 s.d.) as expected (Fig. S2). To test for synchronous gametocyte production, NF54/iGP1_RE9H^ulg8 ring stage parasites were treated with −GlcN/ + Shield-1 for 48 h to induce sexual commitment and the ring stage progeny (day 1 of gametocyte differentiation) was cultured in medium supplemented with 50 mM GlcNAc for six days to selectively eliminate asexual parasites[32,33] and in normal culture medium thereafter. Based on microscopic inspection of Hemacolor-stained blood smears, NF54/iGP1_RE9H^ulg8 gametocytes displayed typical morphology throughout development, gametocyte maturation was highly synchronous, and on day 12 virtually all gametocytes displayed stage V morphology (92.5% ± 4.5 s.d.) (Fig. 1B and Fig. S2). Starting with a ring stage parasitemia of 1–2 % in the induction cycle routinely delivered day 12 gametocyte cultures at 2–5% gametocytemia (3.6% ± 1.9% s.d.), making further purification or enrichment of gametocytes unnecessary. Furthermore, immunofluorescence

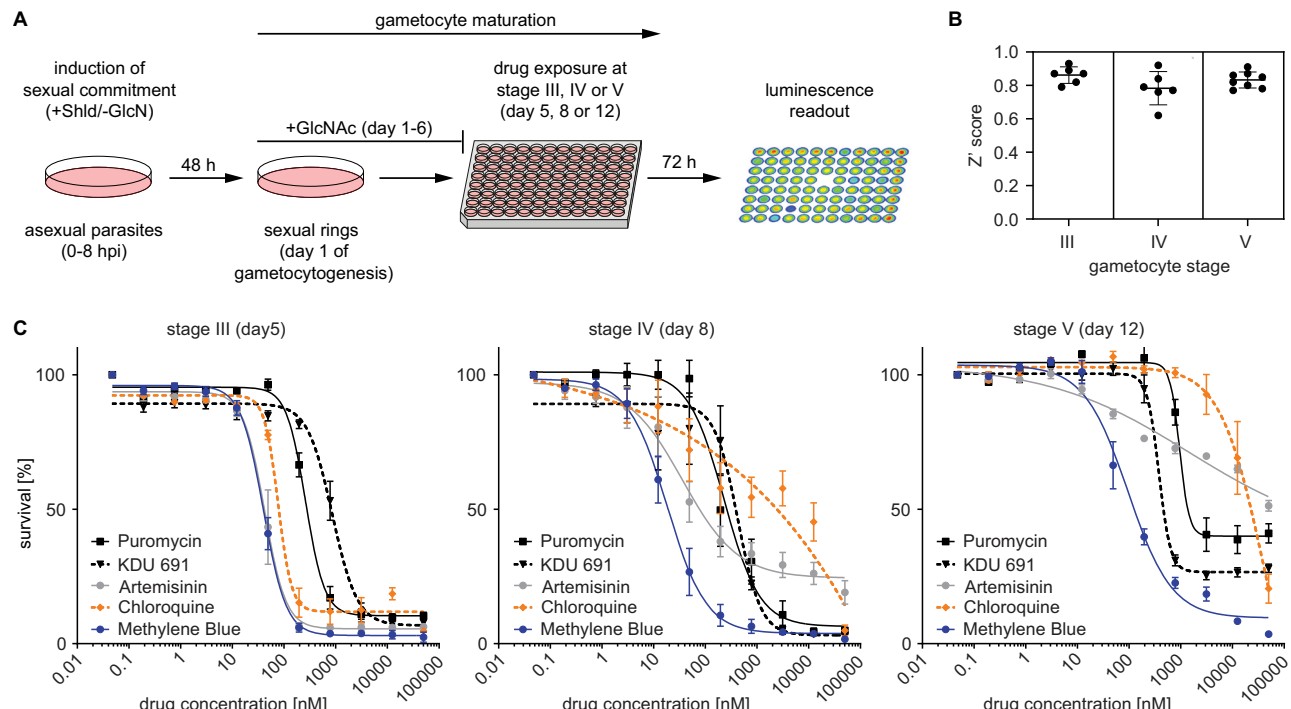

**Fig. 2 | Development of an in vitro gametocytocidal drug assay using NF54/iGP1_RE9H<sup>ulg8</sup> gametocytes.** Assay setup and validation. **A** Schematic illustrating the workflow used to produce synchronous NF54/iGP1_RE9H<sup>ulg8</sup> gametocytes and stage-specific gametocytocidal compound testing in a 96-well plate format. Gametocytes are exposed to compounds at the desired day of gametocyte development and gametocyte viability is determined 72 h later via RE9H-catalysed bioluminescence readout. Shld, Shield-1; GlcN, glucosamine, GlcNAc, N-acetylglucosamine. **B** Z' scores calculated from the RLUs obtained from four untreated (0.1% DMSO) and four treated (50 µM MB) samples (technical replicates) routinely included on each drug assay plate. Thick horizontal lines represent the

mean Z' scores calculated from $n = 6$ (stage III and IV) or $n = 8$ (stage V) different assay plates, error bars indicate the s.d. and individual values are represented as black dots. **C** Dose-response curves for reference antimalarials (ART, CQ) and experimental compounds (KDU691, puromycin, MB) tested against synchronous NF54/iGP1_RE9H<sup>ulg8</sup> gametocytes at three different stages of development. Values on the y-axis represent RLUs normalized to the mean signal emitted from cells exposed to the lowest drug concentration, obtained from $n = 3$ (stage III and IV) or $n = 4$ (stage V) biological replicates (mean ± s.e.m.). IC$_{50}$ values are shown in Table S1.

---

assays using antibodies against the female-specific marker Pfg377[84] revealed the expected female-biased sex ratio, and gamete activation assays demonstrated that male NF54/iGP1_RE9H<sup>ulg8</sup> stage V gametocytes exflagellated at rates comparable to those observed for NF54 wild type parasites (Fig. S2).

To verify and quantify expression of the RE9H luciferase in live cells, NF54/iGP1_RE9H<sup>ulg8</sup> day 12 gametocyte suspensions were serially diluted at constant hematocrit, incubated with D-luciferin (30 mg/ml) in black-wall 96-well assay plates under non-lysing conditions and bioluminescence was quantified using a camera-based detection system (IVIS Lumina II). Relative luminescence units (RLU) were highly correlated with absolute gametocyte numbers over a wide range (~1320-675,000 gametocytes/well) (Fig. 1C). A serial dilution experiment comparing day 12 gametocytes incubated for 72 h in the presence or absence of the gametocytocidal compound methylene blue (MB) (50 µM) revealed low background luminescence for MB-treated cells and excellent signal-to-background (S/B) and signal-to-noise (S/N) ratios across different gametocyte densities (~10,500–337,500 gametocytes/well; S/B = 3.2–57; S/N = 16–1276) (Fig. S2). Together, these results demonstrate that the RE9H reporter serves as an accurate and sensitive marker to quantify NF54/iGP1_RE9H<sup>ulg8</sup> gametocyte viability.

#### Validation of the NF54/iGP1_RE9H<sup>ulg8</sup> line for in vitro stage-specific gametocytocidal drug activity profiling

To evaluate the suitability of NF54/iGP1_RE9H<sup>ulg8</sup> gametocytes for drug activity profiling and library screening purposes, we established a whole cell-based viability assay. Briefly, ring stage parasites are used as the starting material for the production of pure synchronous

gametocyte populations according to the protocol explained above. Gametocytes are then exposed to test compounds at the desired stage of gametocyte development in 96-well cell culture plates (150 µl final volume; approx. 2% gametocytemia, 1.5% hematocrit, 450,000 gametocytes/well). After 72 h of drug exposure, 90 µl culture suspension is transferred to the wells of black-wall plates preloaded with 10 µl D-luciferin and gametocyte viability is assessed by quantifying RE9H-catalyzed luminescence (Fig. 2A).

To determine assay performance, we exposed NF54/iGP1_RE9H<sup>ulg8</sup> gametocytes at stage III (day 5), stage IV (day 8) and mature stage V (day 12) to antimalarial drugs with known gametocytocidal activity. Dose response assays were performed in $n = 3$ biological replicates (each containing technical duplicates), and each plate included four replicate wells each of gametocytes treated with 50 µM MB (positive control) or the DMSO solvent only (0.1%; negative control) to calculate Z' scores for monitoring assay quality[85]. The luminescence signals measured from the positive and negative control samples on each plate indicated low variability and excellent assay robustness across the three different gametocyte stages tested (Z' score = 0.83 ± 0.04 s.d.) (Fig. 2B). Consistent with published data, chloroquine (CQ) was active only against stage III gametocytes, whereas artemisinin (ART) showed activity against stage III and IV gametocytes but failed to kill stage V gametocytes[6]. In contrast, the phosphatidylinositol 4-kinase (PI4K) inhibitor KDU691[86] and the two control compounds MB and puromycin, all of which are active activity against all gametocyte stages in vitro[6,9], showed the same characteristics in our assay (Fig. 2C, Table S1).

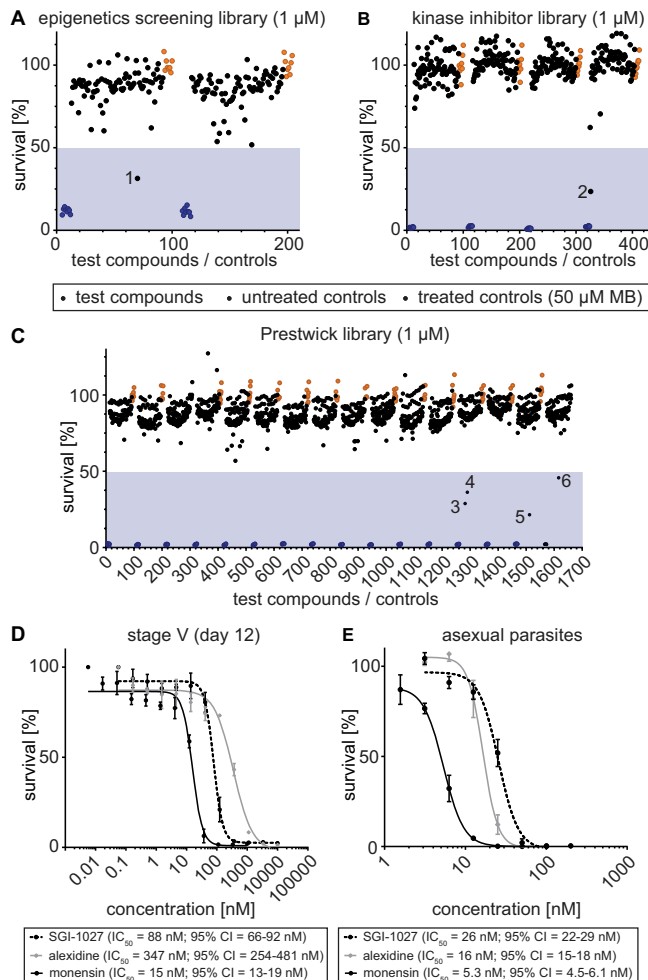

**Fig. 3 | Screening of chemical libraries against NF54/iGP1_RE9H[ulg8] stage V gametocytes.** Results from the primary screening of small- to medium-sized compound libraries. **A–C** Effect of compounds of the Epigenetics Screening Library (Cayman Chemical) (**A**), human kinase inhibitors (SelleckChem, Enzo Life Sciences) (**B**) or compounds of the Prestwick Chemical Library (**C**) on NF54/iGP1_RE9H[ulg8] stage V (day 12) gametocyte viability (1 μM concentration). Each assay plate included eight treated (50 μM MB; blue dots) and untreated (0.1% DMSO; orange dots) samples each as positive and negative controls, respectively. Values on the y-axis represent RLUs normalized to the mean signal emitted from the negative controls, obtained from *n* = 1 experiment for each library. Compounds with >50% inhibitory activity (blue shaded areas) are highlighted by numbers (1, SGI-1027; 2, SU4312; 3, monensin; 4, alexidine dihydrochloride; 5, indoprofen; 6, equilin). **D** Dose-response curves of SGI-1027, monensin and alexidine dihydrochloride tested against NF54/iGP1_RE9H[ulg8] stage V gametocytes (day 12). Values on the y-axis represent RLUs normalized to the mean signal emitted from cells exposed to the lowest drug concentration, obtained from *n* = 3 biological replicates (mean ± s.e.m.) **E** Dose-response curves of SGI-1027, monensin and alexidine dihydrochloride tested against NF54 wild type asexual blood stage parasite multiplication. Values on the y-axis represent [³H]-hypoxanthine incorporation normalized to the mean signal emitted from eight untreated control samples per plate, obtained from *n* = 3 biological replicates (mean ± s.e.m.). IC₅₀ values and 95% confidence intervals (CI) are indicated below the graphs.

## Screening of chemical libraries identifies molecules that kill NF54/iGP1_RE9H[ulg8] mature stage V gametocytes in vitro

To validate this assay as a tool for the discovery of compounds targeting quiescent stage V gametocytes, we screened 1740 molecules from four different chemical libraries. Mature NF54/iGP1_RE9H[ulg8] stage V gametocytes (day 12) were exposed to compounds at two different concentrations (10 μM and 1 μM) for 72 h. Of the 148 molecules represented in the Epigenetics Screening Library (Cayman

Chemical), 19 compounds reduced gametocyte viability by >50% at 10 μM (12.8% hit rate) (Fig. S3, Data file S1). When probed at 1 μM, only a single compound (SGI-1027), a quinoline-based inhibitor of the mammalian CpG-specific DNA methyltransferases (DNMTs) DNMT1 and DNMT3A/3B[87–89], showed greater than 50% inhibitory activity (68.8% inhibition) (0.7% hit rate) (Fig. 3A, Data file S1). Vanheer and colleagues recently screened a related library against *P. falciparum* stage IV/V gametocytes using a mitochondrial membrane potential-dependent fluorescence readout[67]. Among the 101 molecules shared between their and our study, the authors reported six compounds causing >50% inhibition at 1 μM and overall their results correlate well with our data (R² = 0.60), with SGI-1027 identified as the most potent gametocytocidal molecule in both studies (Fig. S4, Data file S2). The slightly higher inhibitory effect of active molecules observed by Vanheer et al.[67] compared to our study may be explained by the fact that our assay uses synchronous stage V gametocytes rather than a mixture of stage IV/V gametocytes (Fig. S4, Data file S2).

We then screened a small focused in-house library consisting of 37 compounds targeting human DNMTs, histone methyltransferases (HKMTs), and deacetylases, several of which have demonstrated fast-acting activity against asexual blood stage parasites[90–95]. Nine of these molecules showed >50% inhibition at 10 μM (21.6% hit rate), but only **68″**/LHER1320, a DNMT3A inhibitor harboring a quinazoline moiety designed to mimic the methyl donor S-adenosyl-L-methionine (SAM)[90], retained >50% inhibitory activity when probed at 1 μM (56.9% inhibition) (2.7% hit rate) (Fig. S3, Data file S3). Interestingly, BIX-01294, another quinazoline-based specific inhibitor of the human G9a HKMT[96] with potent activity against asexual blood stages and male gamete formation[94,95], also showed moderate activity against stage V gametocytes (30% inhibition at 1 μM) (Data files S1 and S3). Similarly active were three quinazoline-quinoline bisubstrate derivatives of **68″**/LHER1320 that carry a quinoline moiety mimicking the targeted cytosine linked to the same (**68**/LH1326) or a differently substituted quinazoline moiety (**20**/LH1281, **A**/LH1514)[90] and were previously shown to kill asexual blood stage parasites with low to mid nanomolar potency[93] (Data file S3).

Next, we screened a library of 275 inhibitors of human kinases (SelleckChem, Enzo Life Sciences). At the 10 μM concentration, eight inhibitors reduced stage V gametocyte viability by >50% (2.9% hit rate) (Fig. S3, Data file S4), and only a single molecule (SU4312), a selective inhibitor of VEGFR2 and PDGFR receptor tyrosine kinases[97], showed >50% inhibitory effect at 1 μM (76.5% inhibition) (0.4% hit rate) (Fig. 3B, Data file S4).

Lastly, we screened the Prestwick Chemical Library containing 1,280 compounds, most of which are FDA- and EMA-approved drugs. The Prestwick Chemical Library has previously been screened against *P. falciparum* asexual blood stage parasites and contains several dozen compounds, including known antimalarials, with >50% inhibitory activity at 5 μM[98]. Here, when tested on mature stage V gametocytes, 40 compounds showed >50% inhibition at 10 μM (3.1% hit rate), and four compounds displayed >50% inhibition at the 1 μM concentration (0.3% hit rate) (Fig. 3C, Fig. S3, Data file S5). These hits are the non-steroidal anti-inflammatory drug indoprofen[99] (78.5% inhibition), the estrogenic steroid equilin[100,101] (54.2% inhibition), the polyether ionophore monensin[102] (71.2% inhibition), and alexidine dihydrochloride[103] (63.2% inhibition), a broad-spectrum antimicrobial[103].

We confirmed the activity of the six most potent compounds identified above through dose-response assays on mature stage V gametocytes and asexual blood stage parasites. All compounds showed a dose-dependent killing effect on stage V gametocytes with IC₅₀ values in the nanomolar range (Fig. 3D, Fig. S5). Three of these compounds (SGI-1027, monensin, alexidine dihydrochloride) demonstrated potent dual-active properties as they also inhibited asexual parasite proliferation with low nanomolar activity (Fig. 3E). In contrast, SU4312, equilin and indoprofen were either inactive or showed

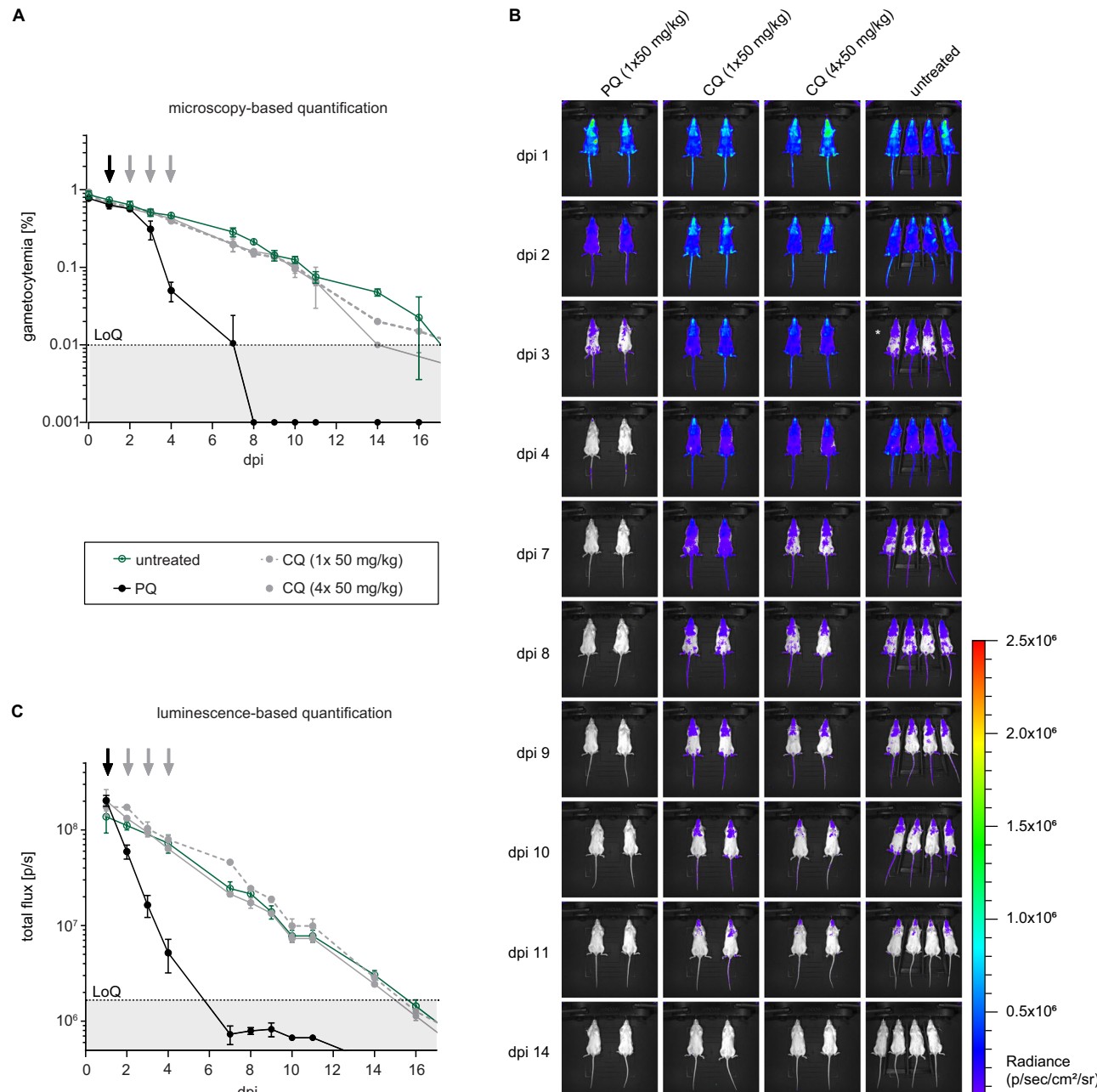

**Fig. 4 | Circulation and clearance of NF54/iGP1_RE9H^ulg8 stage V gametocytes in the NSG-PfGAM in vivo model.** Mice were infected with $2 \times 10^8$ NF54/iGP1_RE9H^ulg8 stage V gametocytes (day 11) and treated one day after infection with $1 \times 50$ mg/kg PQ, $1 \times 50$ mg/kg CQ or four daily doses of $1 \times 50$ mg/kg CQ (days 1–4) or were left untreated. **A** Peripheral gametocytemia in treated and untreated NF54/iGP1_RE9H^ulg8-infected mice as determined by microscopic inspection of Hemacolor-stained thin blood smears prepared daily from tail blood for 16 days. Arrows indicate the day(s) of treatment. Values on the y-axis represent gametocytemia (mean ± s.d.) obtained from $n = 2$ mice per drug/dose combination and $n = 4$ untreated control mice. **B** Representative ventral images of treated and untreated NF54/iGP1_RE9H^ulg8-infected mice. Pseudocolour heat-maps indicate

RE9H-catalysed bioluminescence intensity from low (blue) to high (red). Circulation of NF54/iGP1_RE9H^ulg8 stage V gametocytes was monitored daily for 16 days following infection (only a selection of images is shown). For technical reasons, the uninfected control mice on day 3 post-infection had to be imaged 30 min after D-luciferin injection (instead of 1 min after D-luciferin injection for all other mice), which resulted in reduced signals (white asterisk). **C** Quantification of in vivo bioluminescence emitted from treated and untreated NF54/iGP1_RE9H^ulg8-infected mice. Arrows indicate the day(s) of treatment. Values on the y-axis represent total photon flux (mean ± s.d.) obtained from $n = 2$ mice per drug/dose combination and $n = 4$ untreated control mice. dpi, days post infection; LoQ, limit of quantification (area below the LoQ is shaded gray); p/s, photons/second.

marginal activity against asexual parasites ($IC_{50} > 10$ μM), which suggested these compounds may have specific activity against gametocytes (Fig. S5). However, an in vitro RE9H luciferase inhibition assay using NF54/iGP1_RE9H^ulg8 gametocyte lysates revealed that SU4312 and indoprofen, and to some extent also equilin, directly inhibit RE9H enzyme activity and were therefore considered false-positive hits (Fig. S5).

## Development of a preclinical humanized NODscidIL2Rγ^null mouse model to evaluate stage V gametocytocidal drug activities in vivo

After the successful application of NF54/iGP1_RE9H^ulg8 gametocytes for in vitro drug activity profiling and screening, we anticipated these parasites may also be suitable to develop a preclinical NODscidIL2Rγ^null (NSG) mouse model for evaluation of gametocytocidal and

transmission-blocking drug efficacy in vivo. To this end, NSG mice were engrafted with human RBCs (hRBCs) as previously described[104]. In parallel, synchronous NF54/iGP1_RE9H[ulg8] gametocytes were cultured in vitro according to the protocol explained above and purified by magnetic enrichment using MACS columns. To determine a suitable inoculum for a NSG infection model, hRBC-engrafted mice were infected with either $1 \times 10^7$, $1 \times 10^8$, $2 \times 10^8$ or $1 \times 10^9$ purified day 9 (stage IV/V) or day 11 (stage V) gametocytes via intravenous (i.v.) injection. The circulation of gametocytes in peripheral blood was then monitored daily via microscopic inspection of Hemacolor-stained thin smears prepared from tail blood. Gametocytes circulated in the peripheral blood of all engrafted mice and the initial gametocytemia was positively associated with inoculum size and gradually decreased over time, with circulating gametocytes detectable for more than ten days post-infection in mice infected with >$10^8$ gametocytes (limit of quantification (LoQ): one gametocyte per 10,000 RBCs) (Fig. S6). With an inoculum of $2 \times 10^8$ gametocytes, the average peripheral gametocytemia reached 0.5% ($\pm 0.04$ s.d.) 30 min after infection (Fig. S6). Based on these encouraging results, we tested whether the high gametocyte production rates achieved with the NF54/iGP1_RE9H[ulg8] line in vitro would allow us to infect mice with RBC pellets taken directly from NF54/iGP1_RE9H[ulg8] gametocyte cultures without prior gametocyte enrichment. Indeed, the injection of packed RBCs containing $2 \times 10^8$ gametocytes even increased the peripheral gametocytemia to 0.9% ($\pm 0.07$ s.d.) compared to the 0.5% achieved with MACS-purified gametocytes, and circulating gametocytes were similarly detectable beyond 10 days post-infection (Fig. S6). Lastly, prior to performing whole animal in vivo bioluminescence imaging of NF54/iGP1_RE9H[ulg8] gametocyte-infected NSG mice, we confirmed that i.v. Injection of D-luciferin (150 mg/kg) had no effect on gametocyte densities and circulation times (Fig. S6). Together, these results demonstrate the successful establishment of a reliable humanized NSG mouse model for the exclusive infection with *P. falciparum* stage V gametocytes (NSG-PfGAM).

To determine whether the NSG-PfGAM model can be used to assess stage V gametocytocidal drug activity in vivo, we first tested the antimalarial reference drugs CQ and PQ. Mice were infected with packed RBCs containing $2 \times 10^8$ NF54/iGP1_RE9H[ulg8] stage V gametocytes (day 11) as outlined above. One day after infection, mice were treated by oral gavage with either a single or four daily doses of CQ ($1 \times$ 50 mg/kg, $4 \times$ 50 mg/kg), or with a single dose of PQ ($1 \times$ 50 mg/kg), or were left untreated. In mice treated with CQ, NF54/iGP1_RE9H[ulg8] gametocytes were detectable in circulation until day 14 by both microscopy and in vivo bioluminescence imaging after D-luciferin injection. The decline in gametocytemia and bioluminescence signal over time was identical compared to untreated control mice, regardless of the received CQ dose or treatment regimen (Fig. 4A-C). On the contrary, gametocyte densities in PQ-treated mice began to decline on day two after treatment and reached the LoQ by day 7 when monitored by microscopy (Fig. 4A). The in vivo bioluminescence-based signals for gametocyte viability declined more rapidly already on day 1 after treatment and fell well below the LoQ by day 7 (Fig. 4B-C). These results clearly demonstrate that the NSG-PfGAM model can readily discriminate between drugs that are inactive or active against stage V gametocytes in vivo via the combined bioluminescence and microscopy readouts of gametocyte viability and clearance, respectively, and validate this system as a preclinical model suitable to evaluate antimalarial drugs and drug candidates with regard to their in vivo gametocytocidal properties.

## Using the NSG-PfGAM model to assess stage V gametocyte killing and clearance in vivo after treatment with clinical drug candidates

We then used the NSG-PfGAM model to assess five clinical antimalarial drug candidates that entered phase I (SJ733), phase II (KAE609/

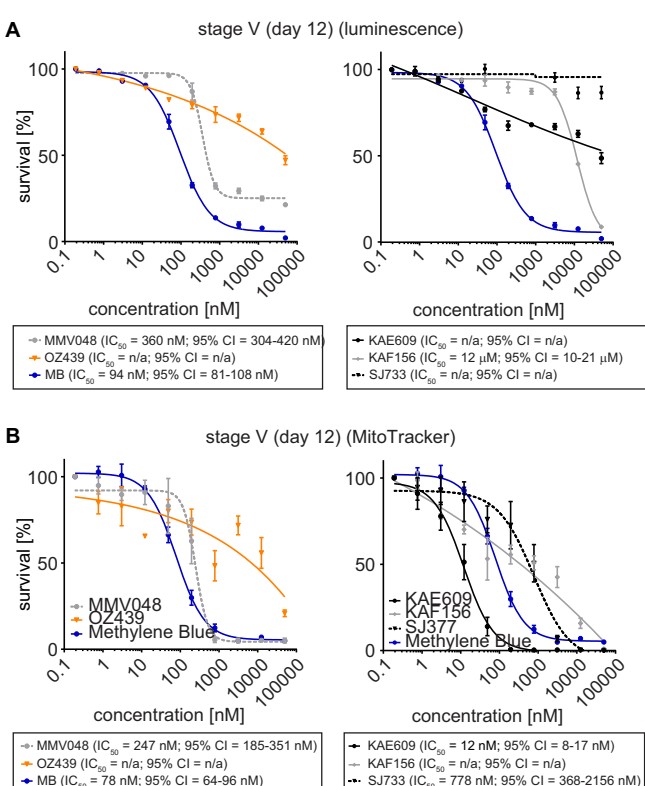

**Fig. 5 | Activities of clinical drug candidates against NF54/iGP1_RE9H[ulg8] stage V gametocytes in vitro.** Comparison of RE9H bioluminescence- and MitoTracker/gametocyte shape-based viability readouts for NF54/iGP1_RE9H[ulg8] stage V gametocytes treated with clinical drug candidates. **A** Dose-response curves of MMV390048 and OZ439/artefenomel (left graph) and KAE609/cipargamin, KAF156/ganaplacide and SJ733 (right graph) tested against NF54/iGP1_RE9H[ulg8] stage V gametocytes (day 12) using RE9H-catalysed bioluminescence as viability readout. MB was used as a positive control (identical data shown in both graphs). Values on the y-axis represent RLUs normalized to the mean signal emitted from cells exposed to the lowest drug concentration, obtained from $n = 3$ biological replicates (mean ± s.e.m.). **B** Dose-response curves of MMV390048 and OZ439/artefenomel (left graph) and KAE609/cipargamin, KAF156/ganaplacide and SJ733 (right graph) tested against NF54/iGP1_RE9H[ulg8] stage V gametocytes (day 12) using MitoTracker signal/gametocyte shape as viability readout. MB has been used as a positive control (identical data shown in both graphs). Values on the y-axis represent normalized mean numbers of viable gametocytes obtained from $n = 3$ biological replicates ($n = 2$ biological replicates for OZ439/artefenomel and MMV390048) (mean ± s.e.m.). $IC_{50}$ values and 95% confidence intervals (CI) are shown below the graphs.

cipargamin, MMV390048, OZ439/artefenomel), and phase III trials (KAF156/ganaplacide) for their activity against stage V gametocytes in vivo. The dihydroisoquinolone SJ733 and the spiroindolone KAE609/cipargamin are chemically unrelated molecules that both target the Na⁺-ATPase PfATP4[105,106]. The imidazolopiperazine KAF156/ganaplacide interferes with protein secretion, but its specific target(s) have not yet conclusively been identified[107,108]. MMV390048 is a PI4K inhibitor[109] and OZ439/artefenomel is a synthetic trioxolane with a peroxide pharmacophore related to that of artemisinin[110]. All five drug candidates have been described as potential transmission-blocking molecules based on in vitro assays, where these compounds showed gametocytocidal activity against mixed late stage gametocytes as assessed by microscopy (KAE609/cipargamin, KAF156/ganaplacide)[107,111] or cellular viability readout (MMV390048)[109], demonstrated inhibitory activity in female (OZ439/artefenomel)[49], male (MMV390048)[109] or dual female/male gamete formation assays (KAE609/cipargamin, KAF156/ganaplacide)[112] or inhibited mosquito infection in SMFAs (KAE609/cipargamin, KAF156/ganaplacide, MMV390048, OZ439/

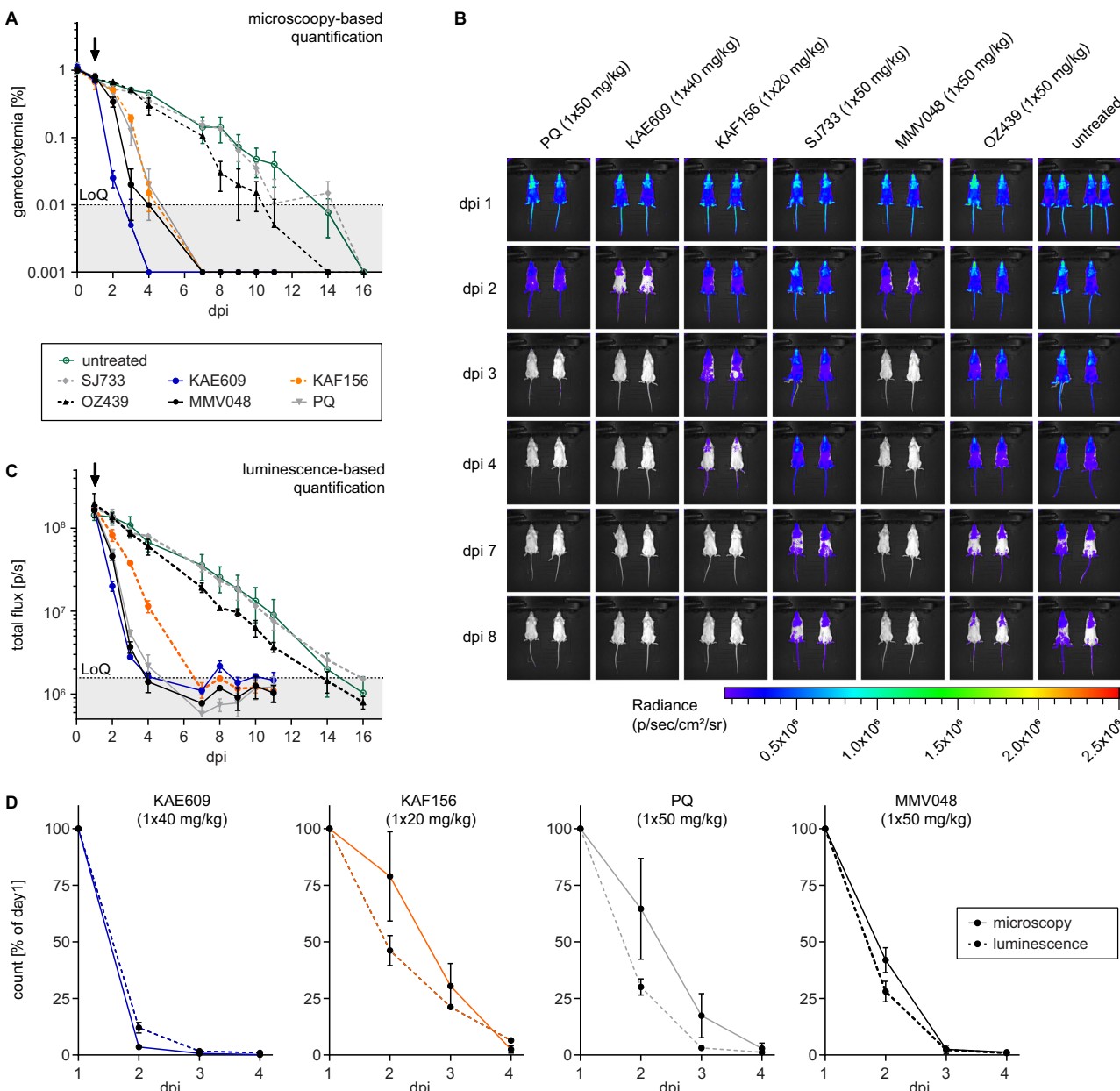

**Fig. 6 | In vivo therapeutic efficacy of clinical drug candidates against NF54/iGP1_RE9H[ulg8] stage V gametocytes in the NSG-PfGAM model.** Mice were infected with $2 \times 10^8$ NF54/iGP1_RE9H[ulg8] stage V gametocytes (day 11) and were treated one day after infection with $1 \times 40$ mg/kg KAE609/cipargamin, $1 \times 20$ mg/kg KAF156/ganaplacide, $1 \times 50$ mg/kg SJ733, $1 \times 50$ mg/kg MMV390048, $1 \times 50$ mg/kg OZ439/artefenomel and $1 \times 50$ mg/kg PQ (positive control) or were left untreated. **A** Peripheral gametocytemia in treated and untreated NF54/iGP1_RE9H[ulg8]-infected mice as determined by microscopic inspection of Hemacolor-stained thin blood smears prepared daily from tail blood for 16 days. The arrow indicates the day of treatment. Values on the y-axis represent gametocytemia obtained from $n = 2$ mice per drug/dose combination and $n = 4$ untreated control mice (mean ± s.d.). **B** Representative ventral images of treated and untreated NF54/iGP1_RE9H[ulg8]-infected mice. Pseudocolour heat-maps indicate RE9H-catalysed bioluminescence intensity from low (blue) to high (red). Circulation of NF54/iGP1_RE9H[ulg8] stage V gametocytes was

monitored for 16 days following infection (only a selection of images up to day 8 is shown). **C** Quantification of in vivo bioluminescence signal emitted from treated and untreated NF54/iGP1_RE9H[ulg8]-infected mice. The arrow indicates the day of treatment. Values on the y-axis represent total photon flux obtained from $n = 2$ mice per drug/dose combination and $n = 4$ untreated control mice (mean ± s.d.). **D** Direct comparison of peripheral gametocytemia (solid lines) and in vivo bioluminescence signal (dashed lines) in NF54/iGP1_RE9H[ulg8]-infected mice treated with active molecules. Values on the y-axis represent % gametocytemia (microscopy readout) or total photon flux (luminescence readout) normalized to the corresponding values determined on day 1 after infection, obtained from $n = 2$ mice per drug/dose combination (mean ± s.d.) (same experiment as shown in panels **A** and **C**). dpi, days post infection; LoQ, limit of quantification (area below the LoQ is shaded gray); p/s, photons/second.

artefenomel)[49,107,109,113,114]. Furthermore, all five clinical drug candidates demonstrated in vivo transmission-blocking activity in the *P. berghei* malaria mouse model[105,107,109,115]. To our knowledge, however, these molecules have never specifically been tested for their in vitro activity against mature stage V gametocytes, except for OZ439/artefenomel that was shown to be inactive[6].

We therefore first performed in vitro dose-response assays to assess the potencies of all five clinical drug candidates against NF54/iGP1_RE9H[ulg8] mature stage V gametocytes (day 12) using the RE9H-based luminescence readout. These experiments confirmed the lack of activity of OZ439/artefenomel and revealed potent activity for MMV390048 ($IC_{50} = 360$ nM) and the positive control MB

(IC$_{50}$ = 94 nM) (Fig. 5A). Surprisingly, stage V gametocyte viability was only poorly compromised by KAE609/cipargamin and KAF156/ganaplacide (IC$_{50}$ = 12 µM) and unaffected by SJ733 (Fig. 5A), even though all compounds potently inhibited asexual parasite proliferation (Fig. S7). However, inspection of Hemacolor-stained thin blood smears revealed strong morphological defects (rounded, swollen gametocytes) upon KAE609/cipargamin exposure as previously observed[111,116], even in the low nanomolar range (Fig. S7). The same phenotype was observed for SJ733-treated gametocytes but only at substantially higher concentrations. KAF156/ganaplacide-treated gametocytes also began to show aberrant morphology at concentrations >200 nM that was clearly distinct from the pyknotic cells observed after exposure to MB (Fig. S7). Hence, KAE609/cipargamin-, SJ733- and KAF156/ganaplacide-treated stage V gametocytes seem to maintain membrane integrity and metabolic activity despite severely compromised cellular morphology. To address this issue further, we employed high content imaging of MitoTracker- and Hoechst-stained cells as an alternative readout to quantify viable stage V gametocytes based on mitochondrial activity and cellular shape. These results uncovered highly potent activity against stage V gametocytes for KAE609/cipargamin (IC$_{50}$ = 12 nM). SJ733 was also active albeit at ~70-fold reduced potency (IC$_{50}$ = 778 nM) compared to KAE609/cipargamin (Fig. 5B). The results for KAF156/ganaplacide were less clear but activity was still apparent even though an IC$_{50}$ value could not be determined. For MMV390048 (IC$_{50}$ = 247 nM), OZ439/artefenomel (inactive) and MB (IC$_{50}$ = 78 nM), the MitoTracker-/cellular shape-based dose response assays delivered highly congruent results compared to the RE9H luciferase-based viability readout (Fig. 5B).

With an improved understanding of their activity against stage V gametocytes in vitro, we tested these clinical drug candidates in vivo in the NSG-PfGAM model. One day after infection with 2 × 10$^8$ NF54/iGP1_RE9H$^{ulg8}$ stage V gametocytes (day 11), mice were treated with either 1 × 40 mg/kg KAE609/cipargamin, 1 × 50 mg/kg SJ733, 1 × 20 mg/kg KAF156/ganaplacide, 1 × 50 mg/kg MMV390048, 1 × 50 mg/kg OZ439/artefenomel, or 1 × 50 mg/kg PQ as a positive control. KAF156/ganaplacide cleared gametocytes by day 4 after dosing with kinetics comparable to those observed after treatment with PQ (Fig. 6A). Furthermore, for both KAF156/ganaplacide and PQ, the bioluminescence readout indicated a faster onset of gametocyte killing compared to clearance, suggesting delayed elimination of dead gametocytes from circulation (Fig. 6B–D). MMV390048 displayed rapid stage V gametocyte killing activity similar to PQ but without any marked delay in gametocyte clearance (Fig. 6A–D). After treatment with OZ439/artefenomel, we observed a lag phase of four days before a slight reduction of gametocytemia and cellular viability could be detected, but circulating gametocytes were still detectable until day 10 and 14 by microscopy and bioluminescence readout, respectively (Fig. 6A–C). Treatment with KAE609/cipargamin resulted in highly rapid clearance of gametocytes to below the LoQ within two days of drug exposure (Fig. 6A–D). Interestingly, the bioluminescence imaging readout highlighted signal accumulation in the upper left quadrant of the abdomen, implying retention and clearance of KAE609/cipargamin-exposed gametocytes in the spleen (Fig. S8). In contrast to KAE609/cipargamin, SJ733 failed to eliminate stage V gametocytes from circulation and gametocytes were detectable by both readouts until day 14 after infection similar to the untreated controls (Fig. 6A–C).

## Using NF54/iGP1_RE9H$^{ulg8}$ gametocyte-infected NODscidIL2Rγ$^{null}$ mice to assess the in vivo transmission-blocking potential of antimalarial drugs and drug candidates

Finally, to test if the NSG-PfGAM model allows assessing in vivo transmission-blocking efficacy of antimalarial drugs and drug candidates, we performed Membrane Feeding Assays (MFAs) using female *Anopheles stephensi* mosquitoes[60,61]. hRBC-engrafted mice were again infected with packed RBCs containing 2 × 10$^8$ NF54/iGP1_RE9H$^{ulg8}$ stage

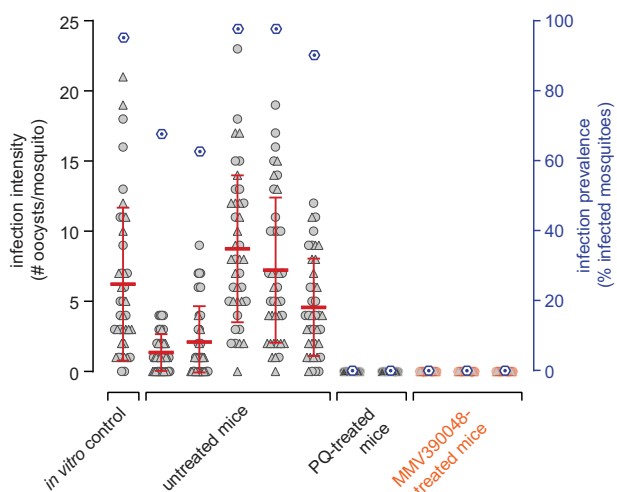

**Fig. 7 | In vivo transmission-blocking efficacy of clinical drug candidates as assessed using the NSG-PfGAM mouse model.** Female *Anopheles stephensi* mosquitoes were allowed to feed on the blood taken from NODscidIL2Rγ$^{null}$ mice infected with 2 × 10$^8$ NF54/iGP1_RE9H$^{ulg8}$ stage V gametocytes (day 11) and were left untreated (positive control for mosquito infection; five mice) or were treated one day after infection with either 1 × 50 mg/kg PQ (positive control for transmission blockade; two mice) or 1 × 50 mg/kg MMV390048 (three mice). Blood samples were prepared for MFAs three days after infection/two days after treatment (day 14 of gametocyte maturation). A day 14 aliquot of the same in vitro gametocyte culture used to infect the mice was prepared for SMFAs and served as a further positive control for mosquito infection. Values on the left y-axis (infection intensity) show the number of oocysts detected in each of the 20 mosquitoes dissected per n = 2 technical replicate feeds, with open circles and triangles representing the results obtained from the two separate feeds, respectively. The thick red horizontal lines represent the mean numbers of oocysts per mosquito, error bars indicate the s.d.. Values on the right y-axis (infection prevalence) represent the mean percentage of infected mosquitoes obtained from the combined replicate feeds for each condition (dark blue hexagons).

V gametocytes (day 11) and treated on the subsequent day with PQ (1 × 50 mg/kg), MMV390048 (1 × 50 mg/kg) or were left untreated. Two days after treatment, whole blood was collected and mosquitoes were allowed to feed on duplicate blood samples of two PQ-treated mice (positive control for transmission blockade), three MMV390048-treated mice, and five untreated control mice (positive control for mosquito infection). Furthermore, a SMFA performed with the same NF54/iGP1_RE9H$^{ulg8}$ gametocyte population used to infect the mice, but maintained in in vitro culture until the day of feeding, served as a positive control for mosquito infection. As shown in Fig. 7, mosquitoes feeding on the blood from untreated mice and the matched in vitro control sample showed a high infection prevalence and comparable infection intensities. In stark contrast, treating mice with a single dose of either PQ or MMV390048 completely blocked gametocyte transmission, with none of the mosquitoes carrying any oocysts (Fig. 7). These results validate the well-established transmission-blocking properties of PQ in this experimental mouse model and reveal that the PI4K inhibitor MMV390048 exhibits similar effectiveness in preventing gametocyte transmission to mosquitoes in vivo.

## Discussion

Here, we engineered the NF54/iGP1_RE9H$^{ulg8}$ reporter line to establish a transmission-blocking drug discovery and development platform that overcomes most challenges associated with conducting in vitro gametocytocidal drug assays and, importantly, provides an urgently needed preclinical model for assessing transmission-blocking drug activity in vivo. NF54/iGP1_RE9H$^{ulg8}$ parasites have two major advantages compared to existing gametocyte reporter lines. First, they are

equipped with an inducible GDV1 over-expression cassette allowing for the routine mass production of synchronous gametocytes[71]. Second, expression of the red-shifted firefly luciferase PpyRE9H not only facilitates quantifying gametocyte viability in vitro but also in vivo via whole animal bioluminescence imaging of NF54/iGP1_RE9H[ulg8]-infected NSG mice. With these two exquisite properties combined, a simple and highly efficient cell culture protocol starting with a 10 ml ring stage culture routinely delivers $2 \times 10^8$ highly synchronous stage V gametocytes that can be directly used without further purification for both in vitro and in vivo drug testing.

For the in vitro assay, gametocytes at the desired stage of maturation are exposed to test compounds in a 96-well format for a defined period of time. At the assay endpoint, the D-luciferin substrate is added just before measuring gametocyte viability using a bioluminescence-based readout. This straightforward assay, requiring only a single well transfer step, shows excellent S/N ratios and high robustness (Z' score > 0.8), making it well-suited for higher throughput formats. While pure synchronous gametocyte populations are obtained as early as day 5 of maturation (stage III), we placed strong emphasis on using mature stage V gametocytes for drug screening as they represent the most difficult-to-kill stages. Of the 1,740 molecules screened against mature stage V gametocytes (day 12), only four compounds (monensin, alexidine dihydrochloride, SGI-1027, 68″/LHER1320) showed >50% activity at 1 μM, which underscores the general insensitivity of these quiescent stages to drug treatment. Importantly, dose-response assays performed for three of these compounds revealed potent dual activity, suggesting they target essential molecular processes shared between asexual parasites and mature stage V gametocytes.

Monensin, as well as other monovalent carboxylic polyether ionophores such as salinomycin and nigericin, are highly active against blood stage parasites in vitro and in vivo[117–120], possess potent gametocytocidal and transmission-blocking activity[63,117] and are interestingly also active against the dormant P. vivax hypnozoite stages[121]. Members of this class of ionophores are also active against other apicomplexan parasites and are widely used in the poultry industry to prevent coccidiosis, an intestinal disease caused by Eimeria spp.[122,123]. Ionophores are lipid-soluble compounds that bind cations and exert their toxicity by disrupting ionic membrane gradients, which in turn affects numerous biological processes[123]. In Toxoplasma gondii, monensin disrupts mitochondrial morphology and membrane potential and elicits oxidative stress, cell cycle arrest, and autophagy-induced cell death[124–126]. While the mechanism of action of monensin against P. falciparum has not been investigated, the profound morphological defects and cell death observed for gametocytes exposed to the dual-active ionophore maduramicin[127] are linked to rapid cellular $Na^+$ influx[128], which interestingly also occurs with the $Na^+$-transporter PfATP4 inhibitors KAE609/cipargamin, SJ733, and PA21A050[105,128–130]. The bis-biguanide alexidine dihydrochloride is a cationic antimicrobial that disrupts bacterial cell wall and membrane integrity and is used as an antiseptic in mouth washes and contact lens solutions[103]. Alexidine dihydrochloride is also active against various pathogenic fungi[131], and its antifungal properties may be linked to inhibition of phospholipases[132]. Further, alexidine dihydrochloride is a specific inhibitor of the human mitochondrial tyrosine phosphatase PTPMT1[133] and was shown to induce apoptosis in cancerous cells in vitro and in vivo[134–136]. Interestingly, alexidine dihydrochloride is listed in the MMV Pandemic Response Box (MMV396785) and was found to be active against Toxoplasma gondii tachyzoites[137] and, consistent with our findings, P. falciparum asexual parasites and late-stage gametocytes[46]. Moreover, alexidine dihydrochloride inhibits male gametogenesis and mosquito infection in both P. falciparum and P. berghei[46,138]. SGI-1027, a non-nucleoside quinoline-based inhibitor of mammalian DNMT1 and DNMT3A/B[87,88], has been proposed to occupy both the cytidine and SAM substrate pockets[139,140] but was then shown

to act via DNA-competitive inhibition and destabilization of the DNMT-DNA-SAM complex[89]. The quinazoline-based compound 68″/LHER1320 is expected to compete with SAM co-factor binding, whereas the moderately gametocytocidal quinazoline-quinoline bisubstrate analogs are predicted to occupy both substrate pockets of DNMTs[90,93]. However, P. falciparum encodes only a single DNMT enzyme, PfDNMT2, that primarily acts as a $tRNA^{ASP}$ methyltransferase[141–143], like its orthologs in other eukaryotes[144], and is not essential for asexual parasite growth, gametocytogenesis and male gamete formation[142,143]. Hence, while SGI-1027 and/or 68″/LHER1320 may still inhibit PfDNMT2 function, their lethal anti-parasitic effects are probably due to inhibition of other unknown vital targets. While HKMTs are obvious candidates to consider, P. falciparum eukaryotic translation initiation factor 3 subunit I has recently been identified as a direct target of compound 70[145], a closely related analog of the bisubstrate inhibitor 68/LH1326 (parent of compound 68″/LHER1320)[90]. In summary, our screening efforts identified and confirmed potent dual-active compounds that may serve as starting points for future structure-activity relationship studies and chemical biology approaches aimed at identifying vulnerable targets in quiescent gametocytes.

Humanized NODscidIL2Rγ[null] (NSG) mice engrafted with hRBCs and infected with P. falciparum serve as invaluable in vivo models for the preclinical development of drugs against asexual blood stage parasites. They are widely employed to test drug candidates for therapeutic efficacy in a physiological environment similar to that of humans, able to capture the effects of drug metabolism and pharmacokinetic properties, determine pharmacodynamic parameters and predict dosing regimens for use in humans[146]. However, these models have never been optimized for assessing drug activity against gametocytes and have, to our knowledge, been used only once to test the gold standard transmission-blocking drug PQ[147], a prodrug that requires metabolic processing to unfold its activity in vivo[148]. Duffier et al. showed that in NSG mice infected with NF54 wild type parasites, four daily intraperitoneal (i.p.) PQ injections (2 mg/kg) cleared the gametocytes that naturally emerged during blood infection within three to six days after treatment initiation[147]. While this study provided important proof of concept that P. falciparum-infected NSG mice can be used to test transmission-blocking drugs, this model requires extensive immunomodulation by repeated i.p. injections, results in mixed infections composed of replicating asexual parasites and varying levels of gametocytes of all stages, and does not support quantifying gametocyte viability in vivo[147]. Here, we addressed these limitations by establishing a NSG model tailored for the systematic evaluation of P. falciparum transmission-blocking drug efficacies in vivo. This NSG-PfGAM model uses a comparatively simple standardized experimental setup, also with regard to animal preparation and handling, thus avoiding any stressful immunomodulation procedures. hRBC-engrafted NSG mice are infected i.v. with 2x10^8 pure NF54/iGP1_RE9H[ulg8] stage V gametocytes (day 11), resulting in a starting gametocytemia of around 1%, and are treated on the subsequent day with drugs/drug candidates administered by oral gavage. Gametocyte viability and clearance can then be monitored and correlated over a period of two weeks by in vivo bioluminescence imaging after D-luciferin injection and microscopic inspection, respectively, and transmission-blocking activity can be assessed by mosquito feeding experiments.

PQ was highly active in the NSG-PfGAM model and cleared stage V gametocytes within four to seven days after treatment, consistent with the findings reported by Duffier and colleagues[147]. Notably, we observed that gametocyte viability declined more rapidly than gametocyte density, demonstrating that PQ-mediated sterilization precedes gametocyte clearance. Indeed, two days of in vivo exposure to PQ rendered gametocytes non-infectious and blocked transmission to mosquitoes. This pharmacodynamic response mimics the gametocyte-sterilizing effect of PQ in human clinical trials, where single-dose PQ

treatments consistently abolish gametocyte infectiousness to mosquitoes by day 2–3 post treatment but a substantial decline in gametocyte densities is only evident by day 7[17,19,149–152]. KAF156/ganaplacide elicited a response similar to that seen with PQ; efficient gametocyte clearance was achieved by day 4 after treatment and the reduction in gametocyte density was preceded by a decline in cellular viability, suggesting that KAF156/ganaplacide also has gametocyte-sterilizing effects. This assumption is consistent with our in vitro data showing that KAF156/ganaplacide-exposed stage V gametocytes retain cellular integrity (despite distorted morphology) even at micromolar concentrations. MMV390048 caused a rapid and coincident decline in both gametocytemia and viability to below the LoQ by day 3 after dosing, and fully blocked transmission to mosquitoes as assessed two days after treatment. Interestingly, MMV390048 also showed promising transmission-blocking potential in recent mosquito infection experiments conducted with stage V gametocytes isolated from naturally infected carriers and exposed to the drug for 24 h ex vivo (IC$_{50}$ ~ 200 nM)[114]. In a recent controlled human malaria infection (CHMI) study, however, single dose treatment of *P. falciparum*-infected volunteers with MMV390048 (80 mg) did not prevent stage V gametocyte formation and mosquito infection[153], indicating higher dosing would be required to efficiently block transmission. OZ439/artefenomel has poor activity against mature gametocytes in cellular viability assays as shown here and elsewhere[6,10,49], but displayed inhibitory effects in vitro on female gamete formation and transmission to mosquitoes with reported IC$_{50}$ values of 100–200 nM[49,113]. In the NSG-PfGAM model, OZ439/artefenomel had no measurable impact on circulating gametocytes during the first three days after treatment, but a moderate reduction in both gametocyte viability and density compared to the untreated controls was observed from day 7 onwards, which may indicate a delayed impact on female and/or male gametocyte fitness. Overall, these results are consistent with those obtained in recent CHMI studies, where single dose treatment with OZ439/artefenomel alone (500 mg) or in combination with DSM265 (200 mg) failed to prevent stage V gametocyte formation and did not reduce their densities five days post treatment[154,155]. While these combined data suggest OZ439/artefenomel lacks in vivo transmission-blocking activity, mosquito feeding assays on treated NSG-PfGAM mice will be required to ultimately confirm this assumption.

Treatment with KAE609/cipargamin resulted in a ~30-fold and ~140-fold reduction in peripheral gametocytemia on the first and second day after dosing, respectively, the fastest in vivo gametocyte clearance rates observed in our study. Notably, this rapid elimination of gametocytes from circulation was linked to accumulation of gametocytes in the spleen. Previous in vitro studies have shown that KAE609/cipargamin inhibits the Na$^+$-ATPase PfATP4 and thereby induces swelling and rigidification of asexual parasite-infected RBCs[156,157] and gametocytes[111,116] (also observed in this study), and that these physical changes prevent their passage through spleen-mimetic filter systems[116,156]. Spleen-dependent elimination of parasitized RBCs has indeed been proposed as the most likely mechanism explaining the rapid clearance of ring stage-infected RBCs in humans after KAE609/cipargamin administration[156,158–160]. Carucci et al. similarly hypothesized that KAE609/cipargamin-exposed stage V gametocytes may also be eliminated in the spleen[116]. The results obtained with our NSG-PfGAM model demonstrate for the first time that KAE609/cipargamin rapidly clears stage V gametocytes from circulation and provide direct in vivo evidence that splenic retention is indeed primarily responsible for this favorable outcome. In clinical studies, however, a single dose of KAE609/cipargamin (10–30 mg) administered to infected study participants was insufficient to prevent the development of gametocytes[159,161], again showing that higher doses may be necessary to achieve effective gametocyte transmission-blocking activity. In stark contrast to KAE609/cipargamin, the second PfATP4 inhibitor we tested, SJ733, had no effect on reducing gametocyte densities throughout the 14 day follow-up period. This profoundly different pharmacodynamic response is likely due to the lower potency SJ733 has against stage V gametocytes in vitro as shown here and in previous SMFA experiments[113,114], combined with inferior pharmacokinetic properties compared to KAE609/cipargamin[105,106]. However, because the bioluminescence-based readout obtained for SJ733-exposed gametocytes only informs about cellular but not functional integrity, we cannot entirely exclude that the gametocytes circulating in treated mice may be attenuated in their capacity to infect mosquitoes.

To date, efficacy assessments of transmission-blocking drug activity in vivo rely on phase 2 clinical trials in naturally infected gametocyte-positive patients participating in mosquito feeding assays before and after treatment; these trials are costly and can only be conducted in a handful of settings with access to specialized laboratories. In recent years, CHMI study protocols have undergone important modifications to enable transmission-blocking drug activity testing in earlier phases of clinical development. Infecting healthy volunteers using the induced blood stage malaria (IBSM) approach[162] followed by piperaquine treatment, which kills asexual parasites but allows gametocytes to develop[163,164], can achieve gametocyte densities high enough to support mosquito infection[154,165] and has recently been used to demonstrate the transmission-reducing effect of single low-dose tafenoquine (50 mg) seven days post-treatment[166]. While the IBSM transmission model represents a major advance in the field, its implementation is again highly complex and expensive and the gametocyte densities obtained are still low and variable such that gametocyte enrichment from venous blood is required to achieve reliable mosquito infection rates[154,165,166]. The preclinical NSG-PfGAM transmission model developed here demonstrated excellent performance and allowed us for the first time to assess and directly compare the in vivo stage V gametocytocidal activities and transmission-blocking potential of several clinical drug candidates. The high starting gametocytemia, long gametocyte circulation times in untreated mice and the combined bioluminescence and microscopy readouts have proven particularly useful for determining and discriminating between the kinetics of gametocyte killing/sterilization and clearance, which highlighted marked differences in the pharmacodynamic responses effected by the different molecules tested. We were thus able to show that after treatment with a single high dose, two of the most advanced clinical drug candidates eliminated stage V gametocytes in vivo with similar (KAF156/ganaplacide) or superior (KAE609/cipargamin) activity compared to the gold standard drug PQ, underscoring their transmission-blocking potential. Furthermore, the NSG-PfGAM model recapitulated the gametocyte-sterilizing effect observed after PQ treatment in humans, indicated that KAF156/ganaplacide may have a similar effect, and revealed that KAE609/cipargamin leads to rapid elimination of stage V gametocytes from circulation, most likely via efficient clearance in the spleen. As previously suggested, the prediction of effective transmission-blocking human doses for drug candidates such as MMV390048, KAE609/cipargamin and KAF156/ganaplacide relies on understanding in vivo drug concentrations over time and their effects on the infectivity of circulating gametocytes[113]. Our NSG-PfGAM transmission model will allow determining in vivo minimum gametocyte-sterilizing concentrations, similar to the minimum parasiticidal concentrations that have been established for compounds active against asexual blood-stage parasites[167], with the proviso that allometric scaling can misestimate pharmacokinetic profiles in humans[168]. We therefore believe that the NSG-PfGAM model, combined with mosquito feeding experiments, offers an invaluable and innovative tool for systematically evaluating transmission-blocking drug candidates and antibodies in vivo under controlled experimental conditions and at an affordable cost. In

addition to early validation of candidate molecules during the pre-clinical phase of development, the NSG-PfGAM model is ideally suited for determining pharmacokinetic/pharmacodynamic relationships and efficacious transmission-blocking doses for clinical candidates, antimalarial drugs and drug combinations as crucial parameters needed to optimally design subsequent clinical studies for efficacy testing in humans.

## Methods

### Plasmodium falciparum in vitro culture

NF54 wild type, NF54/iGP1 and NF54/iGP1_RE9H[ulg8] parasites were cultured using human erythrocytes (Blutspende SRK Zürich, Switzerland; blood groups AB+ or B+) at a hematocrit of 5% in RPMI 1640 medium (10.44 g/l) (Fisher Scientific; #11544506) supplemented with 25 mM HEPES (Roth; #HN77.5), 370 μM hypoxanthine (Sigma-Aldrich; #H9377), 24 mM sodium bicarbonate (Sigma-Aldrich; #S5761), 100 μg/ml neomycin (Sigma-Aldrich; #N6386) and 10% heat-inactivated AB+ human serum (Blutspende SRK Basel, Switzerland). For NF54/iGP1_RE9H[ulg8] asexual cultures, 2.5 mM D-(+)-glucosamine hydrochloride (GlcN) (Sigma-Aldrich; G1514) was routinely added to the growth medium to maintain the glmS ribozyme active. Intra-erythrocytic growth synchronization was performed using repeated sorbitol treatments[169]. Parasite cultures were maintained in an atmosphere of 3% $O_2$, 4% $CO_2$, and 93% $N_2$ in air-tight chambers at 37 °C.

### Cloning of transfection constructs

The pHF_gC-ulg8 CRISPR/Cas9 plasmid was generated by T4 DNA ligase-dependent insertion of annealed complementary oligonucleotides (ulg8_sg1F, ulg8_sg1R) encoding the single guide RNA (sgRNA) target sequence sgt_ulg8 along with compatible single-stranded overhangs into BsaI-digested pHF_gC[72]. The sgt_ulg8 target sequence (ggtcctttatacgacacagg) is positioned 243–223 bp upstream of the ulg8 STOP codon and has been designed using CHOPCHOP[170]. The pD_ulg8_re9h donor plasmid was generated by Gibson assembly[171] of four PCR fragments. The first PCR fragment represented a 523 bp synthetic sequence (GenScript) corresponding to the 3' end of the ulg8 gene, of which the first 275 bp constitute the 5' homology box (HB) and the last 248 bp have been recodonized. PCR fragment 1 was amplified from plasmid pUC57-re-ulg8 (GenScript) using primers 5' box_F and 5' box_R. The second PCR fragment represented a sequence encoding an in-frame fusion of the 2A split peptide and the PpyRE9H luciferase. To generate this template, we inserted a sequence encoding the 2A split peptide directly upstream of the re9h coding sequence in plasmid pTRIX2-RE9h[80]. This was achieved by Gibson assembly of a PCR fragment encoding the 2A sequence, amplified from pSLI-BSD[172] using primers 2a_F and 2a_R, and the BamHI-digested pTRIX2-RE9h plasmid, resulting in plasmid pTRIX2-2A-RE9h. PCR fragment 2 (2A-RE9H) was then amplified from pTRIX2-2A-RE9h using primers 2a-re9h_F and 2a-re9h_R. PCR fragment 3 represented the 830 bp 3' HB spanning the last 226 bp of the ulg8 coding sequence and 604 bp of the downstream sequence (amplified from 3D7 gDNA using primers 3' box_F and 3' box_R). PCR fragment 4 represented the pD donor plasmid backbone amplified from pUC19 using primers PCRA_F and PCRA_R[72]. All oligonucleotide sequences used for cloning are provided in Table S2.

### Transfection and selection of transgenic NF54/iGP1_RE9H[ulg8] parasites

NF54/iGP1 parasites were transfected and selected according to the protocol published by Filarsky and colleagues[72]. RBCs collected from a 5 ml culture of synchronous ring stage parasites (10% parasitemia) were co-transfected with 50 μg each of the pHF_gC-ulg8 CRISPR/Cas9 plasmid and the pD_ulg8_re9h donor plasmid using a Bio-Rad Gene Pulser Xcell Electroporation System (single exponential pulse, 310 V,

250 μF). Twenty hours after transfection, 5 nM WR99210 was added to the culture medium for six days to select for gene-edited parasites, and henceforth parasites were maintained in normal culture medium until a stably propagating parasite population was obtained. The transgenic NF54/iGP1_RE9H[ulg8] line was cloned out by limiting dilution cloning using an established plaque assay[173] and successful editing of the ulg8 locus was confirmed by PCR on gDNA isolated from two clonal lines (A2 and B2). All oligonucleotide sequences provided in Table S2).

### Parasite multiplication rates

Synchronous NF54 wild type and NF54/iGP1_RE9H[ulg8] ring stage cultures were seeded at approx. 1% parasitemia and incubated in culture medium supplemented with 2.5 mM GlcN until completion of one invasion cycle (48 h). Parasitemia at baseline and in the progeny was determined by microscopy (100x immersion oil objective) counting uninfected and at least 200 infected RBCs in thin blood smears stained with Hemacolor Rapid (Merck; #1.11956.2500, #1.11957.2500). Multiplication rates were calculated as the ratio of parasitemia measured in the progeny compared to baseline.

### Induction of sexual commitment and gametocyte culture

Sexual commitment of NF54 wild type parasites was induced by exposing synchronous late ring stage cultures (20–28 h post invasion) to minimal fatty acid medium [culture medium containing 0.39% fatty acid-free BSA (Sigma-Aldrich; #A6003) instead of AlbuMAX II, supplemented with 30 μM oleic acid and 30 μM palmitic acid (Sigma-Aldrich; #O1008 and #P0500)][174]. Sexual commitment of NF54/iGP1_RE9H[ulg8] parasites was triggered as described in Boltryk et al.[71]. Briefly, synchronous ring stage cultures (2–3% parasitemia) were grown for 48 h in medium lacking GlcN and containing 1.25 μM Shield-1 to induce GDV1-GFP-DD expression in trophozoites and schizonts. Eight hours after invasion into new RBCs (8–16 hpi asexual/sexual ring stage progeny; day 1 of gametocytogenesis), the induction medium was replaced with standard culture medium containing 50 mM N-acetylglucosamine (GlcNAc) (Sigma-Aldrich; #A3286) for six consecutive days to selectively kill asexual parasites[32,33]. From day 7 onwards, gametocytes were maintained in standard culture medium. The culture medium was exchanged daily during the first seven days of gametocyte development and subsequently every second day. Gametocytemia and gametocyte morphology was assessed by visual inspection of Hemacolor-stained thin blood smears using a 100x immersion oil objective.

### Exflagellation assays

Exflagellation assays were conducted on NF54 wild type and NF54/iGP1_RE9H[ulg8] mature day 13 gametocyte cultures according to a previously published protocol[28]. In brief, culture suspensions were pelleted at 600 g for 3 min and the RBC pellet was resuspended in activation medium [serum-containing culture medium supplemented with 100 μM xanthurenic acid (Lucerna; #Y-W014666)]. After a 15 min incubation at room temperature, samples were applied to a Neubauer chamber, and exflagellation centers and RBC densities were quantified by bright-field microscopy (Leica DM1000 LED, 40x objective). Gametocytemia was determined by visual inspection of Hemacolor-stained thin blood smears prepared from the same samples. Exflagellation rates were determined as the proportion of gametocytes forming exflagellation centers.

### Immunofluorescence assays

IFAs to visualize Pfg377 expression in NF54 wild type and NF54/iGP1_RE9H[ulg8] late stage gametocytes (day 10) were performed on thin blood smears fixed with ice-cold methanol/acetone (60:40). The slides were incubated for one hour in blocking solution (3% BSA in PBS) followed by one hour incubation with rabbit α-Pfg377 antibodies[84]

(1:1000). After three washes with blocking solution, Alexa Fluor 568-conjugated α-rabbit IgG (Molecular Probes; #A11011) secondary antibodies were applied (1:250) and the slides were incubated for 45 min in the dark. Slides were washed three times in PBS and mounted using Vectashield antifade containing DAPI (Vector Laboratories; #H-1200). A minimum of 200 gametocytes were counted to determine the sex ratio. Microscopy was performed using a Leica Thunder 3D Assay fluorescence microscope (63x objective) equipped with a Leica K5 cMOS camera and Leica Application Suite X software (LAS X version 3.7.5.24914). Identical settings were used for both image acquisition and processing with Fiji (ImageJ2 version 1.54 f).

### Chemical compounds

Primaquine (PQ) (Sigma-Aldrich; #160393), chloroquine (CQ) (Sigma-Aldrich; #C6628), artemisinin (ART) (Sigma-Aldrich; #361593), methylene blue (MB) (Sigma-Aldrich; #M9140), puromycin (Sigma-Aldrich; #P8833), SU4312 (Sigma-Aldrich; #S8567, alexidine dihydrochloride (Sigma-Aldrich; #A8986), monensin (Sigma-Aldrich; #475895), indoprofen (Sigma-Aldrich; #I3132), equilin (Sigma-Aldrich; #E8126), the clinical drug candidates KDU691, KAE609/cipargamin, KAF156/ganaplacide and the experimental antimalarials SJ733, MMV390048, OZ439/artefenomel (all provided by MMV) were prepared as 10 mM stock solutions in 100% DMSO (chloroquine was prepared in ddH$_2$0). The Epigenetics Screening Library (148 compounds) has been purchased from Cayman Chemical (#11076) (Data file S1). The 37 compounds targeting DNMTs, HKMT, and histone deacetylases were synthesized as described[90–93,175,176] (Data file S3). The kinase inhibitor library of 275 compounds (SelleckChem and Enzo Life Sciences) (Data file S4) and the Prestwick Chemical Library of 1280 off-patent small molecules (Data file S5) were provided by the Biomolecular Screening Facility (BSF) (École Polytechnique Fédérale de Lausanne (EPFL), Switzerland). All library compounds were provided as 10 mM stock solutions in 100% DMSO.

### Establishment and validation of a whole cell-based RE9H luciferase assay for in vitro gametocytocidal drug screening

To determine the range within which the absolute number of viable NF54/iGP1_RE9H$^{ulg8}$ gametocytes is linearly correlated with luminescence signals, stage V gametocyte cultures (day 12) were resuspended in culture medium containing 0.5% Albumax II (Gibco; #11021-037) instead of 10% human serum (assay medium) at ~2.5% gametocytemia and 3% hematocrit. 300 μl gametocyte suspension was transferred into the well of a 96-well cell culture plate (Corning; #353072) and serially diluted in assay medium at constant hematocrit in biological quadruplicates, each containing eight technical replicates (150 μl suspension/well; 10-step dilution series; 2-fold serial dilutions; ~1,125,000-2,200 gametocytes/well). To determine signal-to-noise (S/N) and signal-to-background (S/B) ratios, serially diluted stage V gametocyte suspensions (day 12) were prepared in assay medium containing MB (final concentration 50 μM) or the DMSO vehicle alone (final concentration 0.1%) in technical quadruplicates (150 μl suspension/well; 6-step dilution series; 2-fold serial dilutions; ~562,500-17,600 gametocytes/well), followed by a 72-hour incubation period under standard growth conditions. Wells containing uninfected RBCs at 3% hematocrit were used as a further control. To quantify RE9H-catalyzed bioluminescence, 90 μl of each suspension was added to individual wells of a black-wall 96-well plate (Greiner CELLSTAR; #7.655.086) preloaded with 10 μl D-luciferin in PBS (30 mg/ml) (PerkinElmer; #122799). After a 5 min incubation step, luminescence signals were quantified using an IVIS Lumina II in vivo imaging system (Caliper Life Sciences, PerkinElmer) at exposure times varying between 30 sec and 3 min. Data were analyzed and exported to a numerical format (counts) using Living Image (v4.7.2, Perkin Elmer). Luminescence counts of technical replicates were averaged and the data imported into GraphPad Prism (version 8.2.1). S/N and S/B ratios were calculated using the following formulas:

$$S/N = \frac{\text{mean signal(untreated)} - mean\ signal(MB-treated)}{standard\ deviation(MB-treated)} \quad (1)$$

$$S/B = \frac{\text{mean signal(untreated)}}{mean\ signal(MB-treated)} \quad (2)$$

### Dose response assays on gametocytes using the RE9H luciferase-based readout

Dose response assays were performed in biological triplicates, each in technical duplicate. Synchronous NF54/iGP1_RE9H$^{ulg8}$ gametocyte cultures (day 5, day 8 or day 12) were resuspended in assay medium at ~2% gametocytemia and 3% hematocrit. 75 μl gametocyte suspension (~450,000 gametocytes/well) was pipetted into each well of a 96-well cell culture plate, pre-loaded with 75 μl compound serially diluted in assay medium (11-step/12-step dilution series; 3-fold/4-fold serial dilutions). To monitor assay robustness using Z′ scores, each plate included four wells containing MB (final concentration 50 μM) and the DMSO vehicle alone (final concentration 0.1%) as positive and negative controls, respectively. Gametocytes were exposed to the compounds for 72 h under standard culture conditions, followed by the transfer of 90 μl culture suspension to black-wall 96-well plates (Greiner CELL-STAR; #7.655.086) preloaded with 10 μl D-luciferin (30 mg/ml) (PerkinElmer; #122799). Luminescence signals were quantified as explained above. Luminescence counts of technical duplicates were averaged and the data imported into GraphPad Prism (version 8.2.1) and normalized to the plateau of sub-lethal compound concentrations using the "first mean in each data set" function. IC$_{50}$ values were calculated using nonlinear, four parameter (variable slope) curve fitting. Z′ scores were calculated using the luminescence signals emitted from the MB-treated and untreated (DMSO vehicle) control samples. Z′ factor formula:

$$Z' = 1 - \left( \frac{3(\partial DMSO + \partial MB)}{\mu DMSO - \mu MB} \right); \partial = \text{standard deviation};$$

$$\mu = \text{mean}; DMSO = \text{untreated controls}; MB = \text{treated controls} \quad (3)$$

### Dose response assays on gametocytes using the MitoTracker-based readout

Dose response assays were performed in biological triplicates using the same plate setup and compound exposure times as explained above for the RE9H luciferase-based assay using synchronous NF54/iGP1_RE9Hulg8 stage V gametocyte cultures (day 12). MitoTracker-based gametocyte viability was quantified using a modified version of published protocols[6,50]. From each well of the assay plate, 36 μl compound-treated gametocyte suspension was transferred to the wells of a cell culture plate preloaded with 4 μl PBS containing 50 μM MitoTracker Red CMXRos (Invitrogen; #M46752) and 50 μg/ml Hoechst 33342 (Thermo Fisher Scientific; #H3570). Following a 2 hour incubation step at 37 °C, 8 μl of the stained gametocyte suspensions was transferred to the wells of a clear-bottom 96-well high content imaging plate (Greiner; #655090) preloaded with 192 μl PBS (final hematocrit 0.1%). Cells were allowed to settle for 20 min before image acquisition using the ImageXpress Micro XLS widefield high content screening system (Molecular Devices) in combination with the MetaXpress software (version 6.5.4.532, Molecular Devices) and a Sola SE solid state white light engine (Lumencor). Filtersets for the detection of Hoechst (Ex: 377/50 nm, Em: 447/60 nm) and MitoTracker signals (Ex: 543/22 nm, Em: 593/40 nm) were used with exposure times of 80 ms and 100 ms, respectively. Thirty-six sites per well were imaged using a Plan-Apochromat 40x objective (Molecular Devices; #1-6300-0297). Automated image analysis was performed using the

MetaXpress software (see Supplementary Methods) to determine the number of viable gametocytes/well (1294–3572 cells/well imaged for untreated control samples). Two technical replicate wells were imaged, the results averaged and normalized to the plateau of sub-lethal compound concentrations using the "first mean in each data set" function and $IC_{50}$ values were calculated using nonlinear, four parameter (variable slope) curve fitting in GraphPad Prism (version 8.2.1).

### In vitro RE9H luciferase activity assay

To account for potential inhibitory effects of compounds on RE9H luciferase activity, an in vitro RE9H inhibition assay was established. To this end, a culture suspension containing $\sim 2 \times 10^8$ NF54/iGP1_RE9H[ulg8] gametocytes was pelleted at 300 g for 3 min, RBCs were resuspended in 7 ml parasite culture medium and mixed with 7 ml BrightGlo Lysis Buffer (Perkin Elmer; #E2610). Lysate aliquots were snap frozen in liquid nitrogen and stored at −80 °C until use. To test the effect of compounds on RE9H activity, dose response assays were performed in technical duplicates by adding 36 µl parasite lysate to the wells of black-wall 96-well plates (Greiner CELLSTAR; #7.655.086) preloaded with 4 µl of compound serially diluted in BrightGlo Lysis Buffer (6-step dilution series; 5-fold serial dilutions). Luciferase inhibitor II (Merck; #119114) was used as a positive control[69,177]. On each plate, 12 wells lacking compound were used as negative controls. Mixtures were incubated for 60 min at room temperature prior to adding 4 µl of D-luciferin (30 mg/ml) (Perkin Elmer; #122799). After additional incubation for 5 min, luminescence signals were quantified at an exposure time of 60 seconds. Luminescence counts of the technical duplicates were averaged, normalized to the untreated controls and the data imported into GraphPad Prism (version 8.2.1). Dose response was evaluated using nonlinear, four parameter (variable slope) curve fitting.

### In vitro screening of chemical libraries against stage V gametocytes

Primary screens of chemical libraries were performed at compound concentrations of 10 µM and 1 µM in the 96-well plate format. Compounds of the Epigenetics Screening Library (Cayman Chemical; #11076) and the collection of 37 DNMT, HKMTs and histone deacetylase inhibitors were manually pre-loaded (100 µl compound diluted in assay medium at 20 µM or 2 µM). NF54/iGP1_RE9H[ulg8] stage V gametocyte cultures (day 12) were resuspended in assay medium at ~2.5% gametocytemia and 3% hematocrit and 100 µl gametocyte suspension was added to each well. For the kinase inhibitor collection and Prestwick Chemical Library, compounds were spotted as 100 nl and 10 nl droplets at 10 mM in 100% DMSO. 50 µl assay medium was manually added to each well to obtain 20 µM or 2 µM compound concentrations, followed by the addition of 50 µl gametocyte culture suspension (note that the three compounds Prestw-1752, Prestw-410, Prestw-354 of the Prestwick library were not spotted correctly and where therefore not screened). Each screening plate included eight wells each containing MB (final concentration 50 µM) or the DMSO vehicle alone (final concentration 0.1%) as positive and negative controls, respectively. Gametocytes were exposed to the compounds for 72 h under standard culture conditions, followed by the transfer of 90 µl of culture suspension form each well to black-wall 96-well plates (Greiner CELL-STAR; #7.655.086) preloaded with 10 µl of D-luciferin (30 mg/ml) (Perkin Elmer; #122799). Luminescence signals were quantified as explained above. Data were normalized to the mean value obtained from the untreated controls (DMSO vehicle) and imported into GraphPad Prism (version 8.2.1). Hits were selected using a 50% inhibition threshold and confirmed in dose-response experiments as described above.

### Dose response assay on asexual blood stage parasites

*P. falciparum* NF54 wild type parasites cultured in medium containing 0.5% Albumax II instead of 10% human serum were used to test for compound activity on parasite multiplication using a [³H]-hypoxanthine incorporation assay[178]. Compounds were dissolved in DMSO at 10 mM concentration, serial dilutions prepared in hypoxanthine-free culture medium (7-step dilution series; 2-fold/4-fold serial dilutions), and 100 µl aliquots dispensed in duplicates into 96-well cell culture plates. 100 µl asexual parasite culture suspension (prepared in hypoxanthine-free medium) were added to each well and mixed with the preloaded compounds to obtain a final hematocrit of 1.25% and a parasitemia of ~0.3%. Each plate included eight wells containing the DMSO vehicle alone (0.1% final concentration) and four wells containing uninfected RBCs (uRBC). After incubation for 48 h, 0.25 µCi of [³H]-hypoxanthine (CliniScience; #ART-0266-5) was added per well and plates were incubated for an additional 24 h. Parasites were harvested onto glass-fiber filters using a Microbeta FilterMate cell harvester (Perkin Elmer, Waltham, USA) and radioactivity was counted using a MicroBeta2 liquid scintillation counter (Perkin Elmer, Waltham, USA). The results were recorded, processed by subtraction of the mean background signal obtained from the uRBC controls, and expressed as a percentage of the mean signal obtained from the untreated controls. Normalized data from each biological triplicate was imported into GraphPad Prism (version 8.2.1) and $IC_{50}$ values were calculated using nonlinear, four parameter (variable slope) curve fitting.

### Magnet-based purification of gametocytes

Gametocyte-infected erythrocytes were purified using MACS D columns (Miltenyi Biotec; #130-041-201) in combination with a Super-MACS™ II Separator (Miltenyi Biotec). Columns were equilibrated with 3% bovine serum albumin in PBS (phosphate-buffered saline) for 15 min. 180 ml of NF54/iGP1_RE9H[ulg8] gametocyte culture (day 8 or 10) was centrifuged at 300 g and resuspended in 60 ml pre-warmed culture medium lacking human serum or Albumax II (wash medium) before allowing the cells to pass through the MACS column attached to a BD Ultra-Fine™ Original Pen Needle 12.7 mm × 29 G (Becton Dickinson; #328203) at low flow rate (<1 drop/sec). The column was washed with pre-warmed wash medium until the flow through was clear by visual observation. Gametocytes were eluted with 50 ml pre-warmed wash medium after detaching the column from the magnetic stand. Eluted gametocytes were centrifuged at 300 g for 3 min, resuspended in 60 ml pre-warmed culture medium and maintained under standard culture conditions for one additional day. Prior to infection of NSG mice, gametocytes were collected by centrifugation and washed once with RPMI 1640 medium supplemented with 25 mM HEPES and 3.1 mM hypoxanthine. Gametocytaemia was determined microscopically and suspensions containing $10^7$, $10^8$, $2 \times 10^8$ or $10^9$ gametocytes in 200 µl RPMI 1640 medium supplemented with 25 mM HEPES and 3.1 mM hypoxanthine were prepared for mouse infections (see below).

### NODscidIL2Rγ[null] mouse model (NSG-PfGAM) for the in vivo testing of gametocytocidal and transmission-blocking drug candidates

Female NODscidIL2Rγ[null] (NSG) mice (nine weeks old) were engrafted by daily intravenous (i.v.) injection of 0.8 ml AB+ or B+ human RBCs (hRBCs) (60% hematocrit in RPMI 1640 medium supplemented with 25 mM HEPES, 3.1 mM hypoxanthine and 20% heat-inactivated AB+ human serum) for 11 days[104]. Mice were housed in individually ventilated cages with an ambient temperature of 21 °C, relative humidity of 55% and a 12-hour dark/light cycle. In pilot experiments, hRBC-engrafted mice were infected by i.v. injection of magnet-purified NF54/iGP1_RE9H[ulg8] gametocytes (day 9 or 11) prepared as explained in the section above. In subsequent optimization experiments, hRBC-engrafted mice were infected by i.v. injection of $2 \times 10^8$ NF54/iGP1_RE9H[ulg8] stage V gametocytes taken directly from in vitro cultures without prior purification. To this end, 400–570 µl packed RBCs of

~3.5-5% gametocytemia cultures were resuspended in standard culture medium to obtain 800 µl suspensions at 50–70 % hematocrit.

For drug treatment experiments, NSG mice were infected with 2 × 10⁸ NF54/iGP1_RE9H$^{ulg8}$ stage V gametocytes (day 11) and treated one day after infection with either a single (1 × 50 mg/kg) or four daily doses (4 × 50 mg/kg) of chloroquine (CQ) (negative controls), a single dose of primaquine (PQ) (1 × 50 mg/kg) (positive control), or single doses of the clinical drug candidates KAE609 (1 × 40 mg/kg), KAF156 (1 × 20 mg/kg), SJ733 (1 × 50 mg/kg), MMV390048 (1 × 50 mg/kg) or OZ439/artefenomel (1 × 50 mg/kg). All compounds were formulated in 70% Tween 80 ($d = 1.08$ g/ml) and 30% ethanol ($d = 0.81$ g/ml), followed by a 10-fold dilution in ddH$_2$O to achieve the desired concentration and administered by oral gavage. Engraftment with hRBCs was continued after treatment every second day until termination of the experiment. Gametocytemia was followed by daily collection of 2 µl tail blood for up to 16 days and determined by visual inspection of Hemacolor-stained thin blood smears, counting at least 10,000 RBCs (limit of quantification (LoQ) threshold set at 0.01% gametocytemia)[104]. For daily in vivo imaging, mice were injected i.v. with 150 mg/kg D-luciferin (Perkin Elmer; #122799) and anaesthetized with gaseous isoflurane (2.5% (v/v) in oxygen). After one minute, mice were placed in an IVIS Lumina II in vivo imaging system (Caliper Life Sciences, PerkinElmer) and ventral images were acquired at an exposure time of 3 min while continuing anesthesia. To measure gametocyte circulation, regions of interest (ROIs) of 4 × 15 cm were placed around the mice and bioluminescence signals were quantified and expressed as total flux (photons/second) using Living Image (v4.7.2, Perkin Elmer). The numerical data were imported into GraphPad Prism (version 8.2.1). The LoQ threshold for in vivo imaging was estimated using the same ROI from at least two uninfected control mice and from infected control mice that did not receive D-luciferin injection. The LoQ (background bioluminescence) was defined as the [mean +3 s.d.] of bioluminescence signal measured within the ROIs from ventral images of these control mice. After imaging, mice were revived and returned to cages.

### Membrane feeding assays

NSG mice were infected with packed RBCs containing 2 × 10⁸ synchronous NF54/iGP1_RE9H$^{ulg8}$ stage V gametocytes (day 11) and were treated on the subsequent day with 1 × 50 mg/kg PQ (two mice) or 1 × 50 mg/kg MMV390048 (three mice) or were left untreated (five mice) as described above. Two days after treatment, NF54/iGP1_RE9H$^{ulg8}$ stage V gametocytes were harvested from these mice and tested in a Membrane Feeding Assay (MFA)[60,61]. Approximately 1 ml of whole blood was collected from each mouse by retro-orbital extraction into lithium-heparin tubes (Becton Dickinson; #368496) containing 1.5 ml pre-warmed culture medium lacking serum. Gametocytemia was quantified by visual inspection of Hemacolor-stained blood smears (mean gametocytemia in untreated control mice = 0.25% ± 0.05 s.d.). Each sample was split in two and spun for 20 seconds at 10,000 g. Pellets were washed once with 1 ml pre-warmed fetal bovine serum (FBS) (Thermo Fisher Scientific; #10270106) and resuspended in 125 µl pre-warmed FBS for feeding (approx. 65% hematocrit; 0.25% gametocytemia in samples from untreated control mice). An aliquot of the same gametocyte culture used to infect the mice was maintained in culture until the day the mice were sacrificed and used to prepare a gametocyte batch- and age-matched in vitro control sample for the MFA. To this end, two 300 µl aliquots of gametocyte culture suspension (3% hematocrit, 1.4% gametocytemia) each were mixed with 180 µl pre-warmed packed RBCs and the cells were pelleted for 20 sec at 10,000 g. The supernatants were removed and the RBC pellets resuspended in 150 µl pre-warmed FBS (approx. 55% hematocrit; 0.1% gametocytemia). All samples were kept at 37 °C at all times and small membrane feeders were used to feed female *Anopheles stephensi* mosquitoes separately with each of the prepared gametocyte samples. On day 7 after feeding, midguts of 20 mosquitoes per feed were

dissected, stained for oocysts using 1% mercurochrome and the numbers of infected mosquitoes and oocysts per mosquito midgut were enumerated by microscopy (400x magnification).

### NODscidIL2Rγ$^{null}$ mouse studies

Animal studies performed at Swiss TPH were approved by the Veterinary Authorities of the Canton Basel-Stadt (permit no. 2992) based on Swiss cantonal (Verordnung Veterinäramt Basel-Stadt) and national regulations (the Swiss Animal Protection law, Tierschutzgesetz). All animal experiments performed at Radboud University were performed in accordance with the Dutch Experiments on Animals Act (Wod) and Directive 2010/63/EU from the European Union and the European ETS 123 convention, and were approved by the Radboud University Animal Welfare Body (IvD) and Animal Experiment Committee (RUDEC; 2015-0142) and the Central Authority for Scientific Procedures on Animals (CCD; AVD103002016424).

### Data analysis

Primary drug screens were performed using two drug concentrations (1 and 10 µM) with single data points. Hit calling was done using a 50% RLU intensity threshold at 1 µM with the mean of untreated parasite cultures defined as 100%. Unless stated otherwise, in vitro experiments were performed in biological triplicates, each comprising at least two technical replicates. To calculate the dose response of parasites to drugs, replicate values were averaged and normalized to those obtained from either untreated parasites or the plateau of ineffective drug concentrations. Response curves were fitted in GraphPad Prism (version 8.2.1) using a nonlinear, four parameter (variable slope) method with error bars representing the standard error of the mean. Error bars of other in vitro as well as in vivo data represent standard deviations. Illustrations and graphs were compiled using Adobe Illustrator (version 29.5.1).

### Reporting summary

Further information on research design is available in the Nature Portfolio Reporting Summary linked to this article.

## Data availability

All data generated in this study are included in this published article and its Supplementary Information and Data files. Source data are provided with this paper. The Source Data file contains the raw data underlying all graphs presented in the main manuscript and Supplementary Information file. Correspondence and requests for materials should be addressed to T.S.V, M.R. and N.M.B.B. Source data are provided with this paper.

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

## Acknowledgements

We are grateful to Sylwia Boltryk and Matthias Wyss from the Swiss Tropical and Public Health Institute for technical assistance and Julien Bortoli from the Biomolecular Screening Facility at EPFL for compound plating and automation. We also thank the staff of the Radboudumc Central Animal Laboratory at the Radboud University Medical Center, in particular Manon van Hulzen and Christien Frederiks for the help with the permits and import of the humanized mice, Charlene Bender and Denise de Jong for their excellent technical support with handling of mice, Marga van de Vegte-Bolmer for parasite culture and Jolanda Klaassen, Laura Pelser-Posthumus, Astrid Pouwelsen, Saskia Mulder and Jacqueline Kuhnen for mosquito breeding and handling of infected mosquitoes. This work was supported by grants from the Fondation Pasteur Suisse (T.S.V., A.S.), Medicines for Malaria Venture (PO17/00583, M.R.), the Swiss National Science Foundation (BSCGI0_157729, T.S.V.; 310030_200683, N.M.M.B.; NCCR Chemical Biology, M.C., G.T.), The Netherlands Organisation for Scientific Research (864.13.009, T.W.A.K.), the European Research Council (ERC-CoG 864180 QUANTUM, T.B.) and the French Agence Nationale de la Recherche (ANR-2019-ANR-20-CE18-0006 EpiKillMal, P.B.A., A.S.).

## Author contributions

Conceptualization: T.S.V., M.R., N.M.B.B., D.L., B.B. Methodology: T.S.V., M.R., N.M.B.B., C.G., T.W.A.K., B.T.T. Validation: N.M.B.B., C.G., G.J.G. Formal Analysis: N.M.B.B., M.R., T.S.V. Investigation: N.M.B.B., C.G., G.J.G., X.Y., A.P., F.N., B.T.T., L.H. Resources: T.S.V., M.R., N.M.B.B., M.C., G.T., P.B.A., A.S., T.W.A.K., T.B. Data curation: N.M.B.B., M.R., T.S.V. Writing – original draft: T.S.V., M.R., N.M.B.B. Writing – review & editing: T.S.V., M.R., N.M.B.B., C.G., F.N., M.C., G.T., P.B.A., A.S., D.L., B.B., M.D., T.W.A.K., T.B. Visualization: N.M.B.B., M.R., T.S.V. Supervision: T.S.V., M.R., A.S., P.B.A., T.W.A.K., N.M.B.B., T.B. Project administration: T.S.V., M.R., N.M.B.B., T.W.A.K., D.L., B.B., M.D. Funding acquisition: T.S.V., A.S., P.B.A., M.R., N.M.B.B., T.W.A.K., M.C., G.T., T.B.

## Competing interests

D.L. is an employee of Medicines for Malaria Venture (MMV). B.B. and M.D. are employees of the Global Antibiotic Research & Development Partnership (GARDP) and were employed at MMV at the time of conceptualization and execution of the work. The remaining authors declare no competing interests.

## Additional information

[1]Department of Medical Parasitology and Infection Biology, Swiss Tropical and Public Health Institute, 4123 Allschwil, Switzerland. [2]University of Basel, 4001 Basel, Switzerland. [3]Department of Medical Microbiology, Radboudumc Center for Infectious Diseases, Radboud University Medical Center, 6525GA Nijmegen, The Netherlands. [4]Biology of Host-Parasite Interactions, Department of Parasites and Insect Vectors, Institut Pasteur, CNRS EMR9195, INSERM U1201, Université de Paris-Cité, Paris 75015, France. [5]Biomolecular Screening Facility, École Polytechnique Fédérale de Lausanne (EPFL), 1015 Lausanne, Switzerland. [6]Epigenetic Chemical Biology, Department of Structural Biology and Chemistry, Institut Pasteur, CNRS UMR3523 Chem4Life, Université de Paris-Cité, Paris 75015, France. [7]Medicines for Malaria Venture, 1215 Geneva, Switzerland. [8]Present address: Laboratoire de Chimie de Coordination du CNRS UPR8241, LCC-CNRS, Inserm ERL 1289, Université de Toulouse, 31077 Toulouse, France. [9]Present address: Global Antibiotic Research and Development Partnership (GARDP), 1202 Geneva, Switzerland. [10]Present address: Department of Immunology and Infectious Diseases, Harvard T. H. Chan School of Public Health, Boston, MA 02115, USA. ✉e-mail: nicolas.brancucci@swisstph.ch; matthias.rottmann@swisstph.ch; till.voss@swisstph.ch

