## [Peer Review file · Nature Communications]

An all-in-one pipeline for the in vitro discovery and in vivo testing of Plasmodium falciparum malaria transmission blocking drugs

Corresponding Author: Professor Till Voss

Version 0:

Reviewer comments:

Reviewer #1

(Remarks to the Author)

Overall summary

Inhibition of human-to-mosquito transmission of *P. falciparum* is essential for malaria control. To this end, the development of effective antimalarial drugs against stage V gametocytes is desired. However, it has been difficult to use large quantities of pure and homogeneous stage V gametocytes in previous drug assay systems, and the application of reporter proteins, such as luciferase, has been limited. In the present study, the authors developed a very robust in vitro anti-gametocyte drug assay platform for stage V gametocytes by combining a method for the conditioned mass production of gametocytes with stage-specific reporter expression. Furthermore, this assay system was expanded to an in vivo assay system using humanised mice. Overall, the study is of high quality and very convincing. However, there are a few minor comments that can be considered by the authors. Addressing these comments strengthens the conclusions of this study.

(Comments)

1. There are missing citations on reporter-expressing parasites with luciferase as a readout. The following article should be cited in the relevant section on page 5.

(Mogollon CM et al. 2019, PMID: 31058097, Miyazaki Y et al. 2023, PMID: 37438491, Vaughan AM et al. 2012, PMID: 23107927)

2. Why did the authors choose ULG8 for the gametocyte-specific expression? What are the advantages of using the ULG8 promoter over the Pfs16, gexp02, and ap2-g promoters used in previous studies? The rationale for choosing ULG8 should be provided in the manuscript.

3. Is ULG8 expressed in gametocytes of both sexes, male or female? It is recommended that previous findings are presented or experimentally demonstrated whether luciferase expression is sex specific or not.

4 The generated ULG8 reporter line encompasses the T2A skip peptide and luciferase downstream of ULG8. However, under certain conditions, the T2A skip peptide may not function and a fusion protein is expressed. There is concern that luciferase-fused ULG8 may affect gametocyte formation and infectivity to *Anopheles* mosquitoes. To address these concerns, it is recommended that gametocyte formation and mosquito infection potential of the ULG8 reporter line be briefly mentioned in the figures or text as equivalent to the parental strain.

5. It is recommended to briefly mention in the figure or text that the ULG8 reporter line has the same ability to proliferate at the erythrocyte stage as the parental strain.

Reviewer #2

(Remarks to the Author)

This is a really nice advance on the Author's previous paper in Nature Comms (<https://doi.org/10.1038/s41467-021-24954-4>).

In the previous paper, they showed that conditional overexpression of a "sexual commitment factor" (GDV1) by a genetically engineered *P. falciparum* NF54/iGP1 (inducible Gametocyte Producer) strain led to over production of synchronized gametocytes. By culturing NF54/iGP1 long enough after induction, they could get 85% or so Stage V gametocytes. This is important, because stage V gametocytes are the circulating form, in vivo, that mosquitoes take up and are infected with. It is very difficult, in non-engineered systems, to achieve such high mature (Stage V) gametocyte production. In this paper, they further engineer NF54/iGP1 to express a red-shifted luciferase specifically in gametocytes (NF54/iGP1-RE9Hulg8). The red shifted luciferase allows a gametocyte viability assay and they used it to screen stage V gametocytes with 4 different chemical libraries, yielding validated hits of known stageV active compounds, but actually found a dozen or so leads, and they confirmed the six most active in dose response against StageV Gametocytes as well as RBC proliferation forms.

They went on to use Nod/SCID/gammal2R (NSG) KO mice to inject Stage V gametocytes IV, and found they were detectable for more than 10d post injection by IVIS. They were able to demonstrate primaquine, but not chloroquine, led to reductions of Stage V gametocytes shortly after administration. They then went on to use the model to evaluate 5 clinical antimalarial candidates that went on to Phase 1-II-III trials. Disappointingly, KAE609/cipargamin, KAF156/ganaplacide, and SJ733 did not appear to potently reduce Stage V parasites, though the compounds were quite active against RBC forms. StageV morphology was abnormal after KAE609/cipargamin and SJ733 treatment. MMV390048 did rapidly reduce stageV gametocytes and also, along with Primaquine, was found to reduce infectivity to mosquitoes. Thus the authors have established a complete pipeline to monitor anti-gametocyte stage V active drugs, and it is a significant advance from existing ways to get Stage V gametocytes (described well in the introduction) for this important drug testing to establish transmission blocking activity.

1. Is it possible that NF54/iGP1-RE9Hulg8 derived gametocytes are any different than naturally occurring gametocytes. Does the overexpression of GDV1 affect mRNA/protein levels of other genes that could affect sensitivity to antimalarial drugs?

2. Can the investigators detect an anti-male or anti-female Stage V gametocyte active compound?

Reviewer #3

(Remarks to the Author)

The work builds upon stage V gametocytes expressing a red-shifted firefly luciferase as reporter which are viable in SCID mice and able to transmit to mosquitoes. They use a GDV1 knockdown to increase gametocytes to about 8% production more than 10 times normal. The work is a tour de force and will be a relevant contribution. The oocyst assay is a must in the pipeline and where other published works fell short of using the oocyst standard. Stage v gametocytes can look live or dead and still produce oocysts.

The work starts out with morphologic changes observed by 5 advanced pipeline drugs from MMV with complicated advances assays to attempt to ascertain viability. They screened hundreds of compounds with some false positives. They use the Prestwick collection and did not cite a large screen of stage IV-V gametocytes with drug exposure on day 15 of gametocyte induction. The compound identified did not overlap knowing the stages were different. (<https://pmc.ncbi.nlm.nih.gov/articles/PMC4144897/>).

They moved to NSG (SCID mouse model with reporter. They noted a strong decrease in signal equated with clearance or killing with different lag phases of reduction. They tested only one new compound with mosquito feeds with primaquine as pos control and no drug control as negative control. This is low number in the gold standard oocyst determination and should be listed as a minor limitation. A greater number of oocyst counts to diverse drugs and decrease in blood stage V gametocytes would have made communication stronger.

80 mg of MMV390048 in humans was minimally effective in human volunteers on mosquito feeds is about 1.3 mg/kg human and equates to near 15 mg/kg in mouse. The 50 mg/kg in mice is then about 3.3 times dose used in humans. Although caveat is to perform the PK as in minority of instances the allometric scaling does not hold up. This should be mentioned more in discussion. (<https://www.fda.gov/media/72309/download>) Also- Freireich, EJ, et al. Quantitative comparison of toxicity of anticancer agents in mouse, rat, dog, monkey and man. *Cancer Chemother Rep.* 1966;50(4):219-244.

The work is a proof of concept but needs more drugs tested in number of oocysts in mosquitos.

Minor comments

Would splenectomy increase circulation time or did they see accumulation of gametocytes in spleen or liver. Images were small but also seen to reflect an increase in intensity in head region in untreated mice. Any ideas

Version 1:

Reviewer comments:

Reviewer #1

(Remarks to the Author)

The authors have responded appropriately to the comments, and I believe that this paper is worthy of publication.

Author response letter

We thank all reviewers for their fair and critical assessment and important input on our manuscript.

Reviewer #1 (Remarks to the Author):

Overall summary

Inhibition of human-to-mosquito transmission of *P. falciparum* is essential for malaria control. To this end, the development of effective antimalarial drugs against stage V gametocytes is desired. However, it has been difficult to use large quantities of pure and homogeneous stage V gametocytes in previous drug assay systems, and the application of reporter proteins, such as luciferase, has been limited. In the present study, the authors developed a very robust in vitro anti-gametocyte drug assay platform for stage V gametocytes by combining a method for the conditioned mass production of gametocytes with stage-specific reporter expression. Furthermore, this assay system was expanded to an in vivo assay system using humanised mice. Overall, the study is of high quality and very convincing. However, there are a few minor comments that can be considered by the authors. Addressing these comments strengthens the conclusions of this study.

>>>We thank reviewer 1 for her/his appreciation of our work.

(Comments)

1. There are missing citations on reporter-expressing parasites with luciferase as a readout. The following article should be cited in the relevant section on page 5.

(Mogollon CM et al. 2019, PMID: 31058097, Miyazaki Y et al. 2023, PMID: 37438491, Vaughan AM et al. 2012, PMID: 23107927)

>>>We apologize for this omission, and we now cite these important articles in the relevant section on page 5 (REFs 55-57).

2. Why did the authors choose ULG8 for the gametocyte-specific expression? What are the advantages of using the ULG8 promoter over the Pfs16, gexp02, and ap2-g promoters used in previous studies? The rationale for choosing ULG8 should be provided in the manuscript.

>>>AP2-G is only expressed in sexually committed schizonts, sexual ring stages and early stage I gametocytes (Kafsack et al., 2014, PMID: 24572369; Bancells et al., 2018, PMID: 30478296) and hence unsuitable as a reporter for late gametocytes. GEXP02 may be an excellent reporter for all gametocyte stages (Portugaliza et al., 2019, PMID: 31601834; Warncke et al., 2020, PMID: 31652487) but had not yet been validated as such at the time when we began to engineer our NF54/iGP1-based reporter line in 2017.

Pfs16 was indeed our first choice and we initially tagged the endogenous *pfs16* gene at its 3' end with the *2a-re9h* fusion sequence (NF54/iGP1_RE9H^{pfs16}) (see Figure 1 below).

Unfortunately, however, these parasites were completely unable to exflagellate (zero exflagellation centers observed) and therefore unsuitable for transmission-blocking drug research. While we did not further investigate this defect, we believe the 2A peptide sequence appended to the C-terminus of Pfs16 may likely be responsible for the lack of exflagellation, as Pfs16 is known to play an essential role in this process (Kongkasuriyachai et al., 2004, PMID: 14698439; Yahiya et al., 2023, PMID: 36715290). (please note that we would prefer not to include these negative results in the manuscript as they are incomplete (e.g. lacking triplicate exflagellation experiments with proper quantification) and in our view don't provide added value, except maybe for the circumstantial but unconfirmed evidence that tagging Pfs16 at the C-terminus may prevent exflagellation.)

Figure 1: Generation of NF54/iGP1_RE9H^{pfs16} parasites. (Top) Schematic map of the endogenous *pfs16* locus, the donor and CRISPR/Cas9 plasmids, and the edited *pfs16* locus expressing a Pfs16-2A-RE9H fusion. Primer binding sites used to confirm successful gene editing by PCR on gDNA are indicated. (Bottom) Results of diagnostic PCRs performed on gDNA from two NF54/iGP1_RE9H^{pfs16} clones (A6 and N6) and NF54 wild type parasites. Note that amplification of the *pfs16* wild type/edited locus using primer pair 1 was very inefficient (faint band visible for NF54 wild type parasites; no band visible for the two NF54/iGP1_RE9H^{pfs16} clones A6 and N6). However, PCRs 2 and 3 clearly demonstrate successful editing of the *pfs16* gene in both NF54/iGP1_RE9H^{pfs16} clones but not in wild type parasites.

Hence, we decided to use ULG8 as a gametocyte-specific reporter based on the results published by Siciliano and colleagues, who demonstrated that both episomal and chromosomally integrated *ulg8* regulatory 5' and 3' regions drive robust expression of GFP and luciferase reporters in both female and male gametocytes, with increased expression observed in late stage gametocytes (Siciliano et al., 2017, PMID: 28118506). We now inserted a corresponding sentence explaining our choice of ULG8 as a gametocyte-specific reporter in the Results section (page 7).

3. Is ULG8 expressed in gametocytes of both sexes, male or female? It is recommended that previous findings are presented or experimentally demonstrated whether luciferase expression is sex specific or not.

>>> To confirm that the RE9H luciferase is indeed expressed in both female and male NF54/iGP1_RE9H^{ulg8} gametocytes, we now performed IFA experiments using antibodies against firefly luciferase. We tested three different commercially available antibodies at different concentrations, raised in three different animals (goat anti-firefly luciferase, NB100-1677, Novus Biologicals; rabbit anti-firefly luciferase, L0159, Sigma-Aldrich; mouse anti-firefly luciferase, L2164, Sigma-Aldrich), and on both methanol/acetone-fixed and formaldehyde/glutaraldehyde-fixed samples. Unfortunately, and very frustratingly, all three antibodies cross-reacted with an unknown parasite protein(s) in NF54 wild type parasites (see Figure 2 below), such that our efforts to confirm the expected expression of the RE9H luciferase in both female and male NF54/iGP1_RE9H^{ulg8} gametocytes were unsuccessful. However, as mentioned above, Siciliano and colleagues already demonstrated that expression of GFP under the control of *ulg8* regulatory 5' and 3' regions occurs in both female and male gametocytes (Siciliano et al., 2017, PMID: 28118506), and we now state this explicitly in the Results section (page 7).

Figure 2: Results of IFA experiments performed on formaldehyde-fixed NF54/iGP1_RE9H^{ulg8} (top two rows) and NF54 wild type (bottom row) gametocytes at day 10 of maturation using three different commercially available antibodies against firefly luciferase.

4. The generated ULG8 reporter line encompasses the T2A skip peptide and luciferase downstream of ULG8. However, under certain conditions, the T2A skip peptide may not function and a fusion protein is expressed. There is concern that luciferase-fused ULG8 may affect gametocyte formation and infectivity to *Anopheles* mosquitoes. To address these concerns, it is recommended that gametocyte formation and mosquito infection potential of the ULG8 reporter line be briefly mentioned in the figures or text as equivalent to the parental strain.

>>> We agree with reviewer 1 that the T2A split peptide may not always function properly. However, even if this was the case with our parasite line, we never observed any obvious differences in gametocyte formation between NF54/iGP1_RE9H^{ulg8} and the NF54/iGP1 parent or NF54 wild type parasites. We now quantified these parameters and demonstrate that NF54/iGP1_RE9H^{ulg8} parasites show (1) normal multiplication rates (see also below); (2) no discernable differences in gametocyte morphology and maturation time; and (3) unaltered gametocyte sex ratios [based on IFA experiments using antibodies against the female-specific Pfg377 protein (Alano et al., 1995, PMID: 8719156; de Koning-Ward et al., 2008, PMID: 18086189)]. Furthermore, as a proxy for gametocyte infectiousness to mosquitoes, we now demonstrate that NF54/iGP1_RE9H^{ulg8} stage V gametocytes exhibit similar exflagellation rates compared to NF54 wild type parasites. These new data are now presented in the Results section (page 8) and in the revised Fig. S2 (panels A-D), and we described the corresponding methods in the Methods section.

5. It is recommended to briefly mention in the figure or text that the ULG8 reporter line has the same ability to proliferate at the erythrocyte stage as the parental strain.

>>>NF54/iGP1_RE9H^{ulg8} parasites never showed any signs of reduced proliferation capacity. We now quantified parasite multiplication rates to confirm that NF54/iGP1_RE9H^{ulg8} replicate as efficiently as NF54 wild type parasites (Results section page 8 and Fig. S2).

Reviewer #2 (Remarks to the Author):

This is a really nice advance on the Author's previous paper in Nature Comms (<https://doi.org/10.1038/s41467-021-24954-4>).

In the previous paper, they showed that conditional overexpression of a "sexual commitment factor" (GDV1) by a genetically engineered *P. falciparum* NF54/iGP1 (inducible Gametocyte Producer) strain led to over production of synchronized gametocytes. By culturing NF54/iGP1 long enough after induction, they could get 85% or so Stage V gametocytes. This is important, because stage V gametocytes are the circulating form, in vivo, that mosquitoes take up and are infected with. It is very difficult, in non-engineered systems, to achieve such high mature (Stage V) gametocyte production.

In this paper, they further engineer NF54/iGP1 to express a red-shifted luciferase specifically in gametocytes (NF54/iGP1-RE9H^{ulg8}). The red shifted luciferase allows a gametocyte viability assay and they used it to screen stage V gametocytes with 4 different chemical libraries, yielding validated hits of known stageV active compounds, but actually found a dozen or so leads, and they confirmed the six most active in dose response against StageV Gametocytes as well as RBC proliferation forms.

They went on to use Nod/SCID/gammaL2R (NSG) KO mice to inject Stage V gametocytes IV, and found they were detectable for more than 10d post injection by IVIS. They were able to demonstrate primaquine, but not chloroquine, led to reductions of Stage V gametocytes shortly after administration. They then went on to use the model to evaluate 5 clinical antimalarial candidates that went on to Phase 1-II-III trials. Disappointingly, KAE609/cipargamin, KAF156/ganaplacide, and SJ733 did not appear to potentially reduce Stage V parasites, though the compounds were quite active against RBC forms. StageV morphology was abnormal after KAE609/cipargamin and SJ733 treatment. MMV390048 did rapidly reduce stageV gametocytes and also, along with Primaquine, was found to reduce infectivity to mosquitoes. Thus the authors have established a complete pipeline to monitor anti-gametocyte stage V active drugs, and it is a significant advance from existing ways to get Stage V gametocytes (described well in the introduction) for this important drug testing to establish transmission blocking activity.

>>>We thank reviewer 2 for her/his appreciation of our work.

1. Is it possible that NF54/iGP1-RE9H^{ulg8} derived gametocytes are any different than naturally occurring gametocytes. Does the overexpression of GDV1 affect mRNA/protein levels of other genes that could affect sensitivity to antimalarial drugs?

>>>In our previous publication detailing the functional analysis of GDV1 (including RNA-seq and ChIP-seq data), we demonstrated that the temporal overexpression of GDV1 has no measurable effect on the expression of genes other than those naturally regulated by GDV1 (*ap2-g* and a few early gametocyte genes) (Filarsky et al., 2018, PMID: 29590075). We also demonstrated that NF54/iGP1 gametocytes (i.e. the parent of NF54/iGP1_RE9H^{ulg8}) undergo gametocyte maturation with normal kinetics and morphology, have unaltered sex ratios, are able to exflagellate, are infectious to mosquitoes, produce oocysts and sporozoites that invade hepatocytes - all with similar efficiency compared to NF54 wild type parasites - showing that overexpression of GDV1 does not affect normal parasite development throughout the life cycle (Boltryk et al., 2021, PMID: 34376675). Furthermore, the drug assay results obtained here with the NF54/iGP1_RE9H^{ulg8} line for antimalarial reference drugs (Fig. 2) and the Epigenetics Screening Library (Fig. S4) are fully congruent with published data obtained with *P. falciparum* wild type parasites or other reporter lines. Lastly, we now performed additional experiments demonstrating that NF54/iGP1_RE9H^{ulg8} parasites show normal multiplication rates, gametocyte development, sex ratios and exflagellation capacity

(Fig. S2A-D). We therefore have no reason to believe that NF54/iGP1_RE9H^{ulg8} gametocytes are any different from NF54 wild type or reporter gametocytes used in other studies.

2. Can the investigators detect an anti-male or anti-female Stage V gametocyte active compound?

>>>No, our assay is not suitable to detect female- or male-specific compound activity. But of course, our NF54/iGP1_RE9H^{ulg8} line can be used in assays tailored to identify compounds with sex-specific activity, such as the dual gamete formation assay (DFGA) (Ruecker et al., 2014, PMID: 25267664).

Reviewer #3 (Remarks to the Author):

The work builds upon stage V gametocytes expressing a red-shifted firefly luciferase as reporter which are viable in SCID mice and able to transmit to mosquitoes. They use a GDV1 knockdown to increase gametocytes to about 8% production more than 10 times normal. The work is a tour de force and will be a relevant contribution. The oocyst assay is a must in the pipeline and where other published works fell short of using the oocyst standard. Stage v gametocytes can look live or dead and still produce oocysts.

>>>We thank reviewer 3 for her/his appreciation of our work.

The work starts out with morphologic changes observed by 5 advanced pipeline drugs from MMV with complicated advances assays to attempt to ascertain viability. They screened hundreds of compounds with some false positives. They use the Prestwick collection and did not cite a large screen of stage IV-V gametocytes with drug exposure on day 15 of gametocyte induction. The compound identified did not overlap knowing the stages were different. (<https://pmc.ncbi.nlm.nih.gov/articles/PMC4144897/>).

>>>We are a bit confused by this comment as the paper mentioned by reviewer 3 (Sanders et al., 2014; PMID: 25157792) did not actually screen the Prestwick library but the Johns Hopkins University Clinical Compound Library v1.3 and the MMV Box, and these authors used mixed late-stage gametocytes for drug screening rather than mature stage V gametocytes. We don't know whether or not the two libraries contain shared compounds. But even if they do, we don't think a comparison of the data would be meaningful given that Sanders et al. and our study used different gametocyte stages for compound screening. However, we thank reviewer 3 for pointing out this paper, which we now cited on page 5 (REF 38).

They moved to NSG (SCID mouse model with reporter. The noted a strong decrease in signal equated with clearance or killing with different lag phases of reduction. They tested only one new compound with mosquito feeds with primaquine as pos control and no drug control as negative control. This is low number in the gold standard oocyst determination and should be listed as a minor limitation. A greater number of oocyst counts to diverse drugs and decrease in blood stage V gametocytes would have made communication stronger.

>>>In a first approach, besides untreated mice we used mice treated with primaquine and chloroquine, known to be active and inactive against stage V gametocytes, respectively, in a proof-of-concept experiment. Furthermore, we tested a selection of five clinical candidates with various modes of action. As outlined on page 13-15, our model revealed interesting new findings in terms of drug efficacy, onset and time until clearance, underscoring the expediency and advantages of this new pipeline. We agree with reviewer 3 that a larger number of compounds would make communication even stronger, but this was out of scope of this study. We envisage using this *in vivo* model for the routine testing of advanced gametocytocidal compounds in the future and are looking forward to testing various new candidate molecules.

80 mg of MMV390048 in humans was minimally effective in human volunteers on mosquito feeds is about 1.3 mg/kg human and equates to near 15 mg/kg in mouse. The 50 mg/kg in mice is then about 3.3 times dose used in humans. Although caveat is to perform the PK as in minority of instances the allometric scaling does not hold up. This should be mentioned more in discussion. (<https://www.fda.gov/media/72309/download>) Also- Freireich, EJ, et al. Quantitative comparison of toxicity of anticancer agents in mouse, rat, dog, monkey and man. *Cancer Chemother Rep.* 1966;50(4):219-244.

>>>The compounds tested in our study were selected from various compound classes with different pharmacodynamic profiles with the intention to illustrate the applicability and usefulness of this new *in vivo* model, and the doses were chosen to enable a comparison between the candidates. The aim of this study was not to determine the PK/PD parameters for various compound classes. But, as pointed out by the reviewer, the presented pipeline was designed to also facilitate exactly such PK/PD studies. The possibility to collect frequent samples for PK analysis and to correlate these results with parallel mosquito feeding assays holds great promise for future studies to better inform the design of clinical studies. We now elaborate more on this aspect in the Discussion section (page 22/23), where we also mention that allometric scaling can misestimate human pharmacokinetic profiles.

The work is a proof of concept but needs more drugs tested in number of oocysts in mosquitos.

>>>As mentioned above, the focus of our work was to establish a drug discovery and preclinical *in vivo* development pipeline for transmission blocking drugs. This proof-of-concept study was not designed to characterize a large number of candidate molecules in more detail but to establish and validate the pipeline with a well-chosen selection of different compounds and clinical candidates with different modes of action to highlight the advantages of this unique *in vivo* model for transmission-blocking drug development.

Minor comments

Would splenectomy increase circulation time or did they see accumulation of gametocytes in spleen or liver. Images were small but also seen to reflect an increase in intensity in head region in untreated mice. Any ideas

>>>We observed no evidence for sequestration of gametocytes in the spleen or liver in untreated mice and can only speculate that splenectomy may increase circulation. We did not perform organ dissection or imaging of isolated organs to further investigate gametocyte sequestration. However, we did find evidence for gametocyte retention in the spleen after treatment with KAE609/cipargamin, as shown in Fig. S8 and discussed on page 21. This observation is consistent with the described mechanism of action of KAE609/cipargamin, which induces rigidification of infected red blood cells. The use of splenectomized mice would certainly be very interesting and informative to further investigate the kinetics of gametocyte elimination and the role of the spleen after treatment with KAE609/cipargamin. The transmission of light through mammalian tissue and skin is influenced by scattering and absorption. In white or hairless mice, light is transmitted more efficiently through the tissue as significant amounts of light are absorbed by melanin. This phenomenon and the naturally extensive blood circulation at snout and mandible in conjunction with the furless skin in this region lead to a strong bioluminescent signal (Li et al., 2017; PMID: 28291500) and is most likely not attributed to sequestration of gametocytes in these areas.